# Protein arginine methyltransferase 5 sustains Tip60-EP400 complex via SRSF1 in Merkel cell carcinoma

Joseph L Sevigny[1,2], Evelyn V Proctor[1] , Heather Adams[1], Erin R MacDonald[1,2], W Kelley Thomas[1,2], Jingwei Cheng[1]

**Protein arginine methyltransferase 5 (PRMT5) is a key regulator of gene expression and RNA splicing, with therapeutic potential demonstrated in MTAP-deleted cancers. Emerging evidence suggests that *MYC*-driven tumors or tumors with wild-type *TP53* may also be sensitive to PRMT5 inhibition, though the underlying mechanisms remain unclear. Virus-positive Merkel cell carcinoma serves as an ideal model to explore this, as it is driven by the *MYC* paralog *MYCL* and retains wild-type *TP53*. In this study, we examined how PRMT5 regulates the Tip60-EP400 complex, which is recruited by MYCL in Merkel cell carcinoma. Using RNA-seq, we characterized PRMT5-mediated gene expression, whereas Iso-Seq enabled in-depth analysis of PRMT5-mediated alternative splicing. How PRMT5 deficiency selectively affects certain splice sites remains unresolved. Our findings suggest that PRMT5-mediated modification of SRSF1 (serine/arginine-rich splicing factor 1) enhances its recruitment to m6A-modified RNA, ensuring proper *KAT5* (Tip60) splicing and Tip60-EP400 activity. PRMT5 inhibition disrupts this recruitment, leading to widespread splicing defects, including exon skipping and intron retention. These results provide new insights into PRMT5's role in splicing regulation and may have broader implications for targeting splicing dysregulation in *MYC*-driven cancers.**

## Introduction

Protein arginine methyltransferase 5 (PRMT5) plays a crucial role in regulating gene expression and RNA splicing by catalyzing symmetrical dimethylation (SDMA) of arginine residues on histone and non-histone proteins. PRMT5 operates within a methylosome complex that includes its core binding partner MEP50 (WDR77) and substrate adaptors such as CLNS1A, RIOK1, and COPR5, fine-tuning its activity to methylate target proteins for specific biological processes (Mulvaney et al, 2021). Its role as a global transcriptional repressor is primarily attributed to the methylation of histone residues H4R3, H3R8, and H2AR3, though it can also facilitate gene

activation through H3R2me2s (Pal et al, 2004; Migliori et al, 2012; Scaglione et al, 2018). The regulatory effects of PRMT5 on chromatin are context-dependent, influencing pathways involved in cell cycle control, differentiation, and apoptosis. PRMT5 represses key tumor suppressors, including *CDKN1A* (p21) and *CDKN2A* (encoding p16INK4a and p14ARF), and modulates p53-responsive gene expression (Wei et al, 2020). In addition, PRMT5 plays a crucial role in maintaining stem cell identity and suppressing apoptotic pathways in proliferative cells (Sloan et al, 2023; DeSisto et al, 2025), though the relative contributions of its histone modification and splicing functions in cancer remain unclear.

Beyond chromatin regulation, PRMT5 plays a canonical role in pre-mRNA splicing by symmetrically dimethylating spliceosomal proteins such as SmB, SmD1, and SmD3. This methylation facilitates their proper loading onto the survival motor neuron (SMN) complex, which aids in the recognition and stabilization of cognate small nuclear RNAs (snRNAs), leading to the assembly of small nuclear ribonucleoproteins (snRNPs). These snRNPs (U1, U2, U4, U5, and U6) are then incorporated into the spliceosome in a stepwise manner to mediate splicing (Meister et al, 2001; Neuenkirchen et al, 2015). The inclusion of CLNS1A (pICln) in the PRMT5 methylosome serves as a checkpoint to ensure splicing fidelity and proper Sm protein integration (Mulvaney et al, 2021). Canonically, PRMT5 influences splicing by modulating snRNP availability, thereby affecting isoform diversity across tissues and disease states such as oncogenesis and neurodegenerative disorders. In glioblastoma, PRMT5 inhibition has been suggested to disrupt detained intron (DI) removal, whereas its effects on canonical alternative splicing, such as exon skipping in tumor growth genes, remain relatively minor (Braun et al, 2017). The broader relevance of PRMT5-mediated intron retention in other cancers, including neuroendocrine tumors, as well as its potential role in exon skipping, remains uncertain. Emerging long-read RNA-sequencing methods suggest that splicing events detected in long-read sequencing are not reliably detected in sample-matched short-read sequencing, highlighting the need for isoform sequencing (Iso-Seq) to capture full-length transcripts and determine PRMT5's impact on splicing (David et al, 2022).

PRMT5 inhibitors have demonstrated synthetic lethality in cancers with MTAP deletions, selectively inducing cancer cell death

---

[1]Department of Molecular, Cellular and Biomedical Sciences, University of New Hampshire, Durham, NH, USA   [2]Hubbard Center for Genome Studies, University of New Hampshire, Durham, NH, USA

Correspondence: jingwei.cheng@gmail.com

(Kryukov et al, 2016; Mavrakis et al, 2016). Sensitivity to PRMT5 inhibition has also been suggested in *MYC*-driven, *TP53* wild-type (WT), and spliceosomal mutant cancers, emphasizing its role in oncogenic splicing (Koh et al, 2015; Li & Diehl, 2015; Lee et al, 2016). In *MYC*-driven tumors, where MYC amplifies RNA production, MYC-induced PRMT5 maintains efficient splicing by preventing exon skipping in transcripts like *EP400*, which is essential for tumor progression (Koh et al, 2015). Virus-positive Merkel cell carcinoma (MCC) serves as an ideal model to explore PRMT5's role in alternative splicing regulation, as it is driven by the *MYC* paralog *MYCL*. Most MCC specimens carry WT *TP53*, and the tumor has a low mutational burden (Starrett et al, 2017), suggesting it is a suitable model for studying alternative splicing. This is supported by an observed inverse correlation between the frequency of alternative splicing changes in a sample and the number of somatic driver mutations (Climente-Gonzalez et al, 2017). MCC is a relatively rare but highly aggressive neuroendocrine tumor of the skin, primarily caused by the integration of Merkel cell polyomavirus (MCPyV) into the host genome and the overexpression of the viral Small T (ST) oncoprotein. Given that neuroendocrine differentiation is associated with poor prognosis and therapy resistance in epithelial cancers (Balanis et al, 2019), elucidating PRMT5's role in MCC may provide insights into splicing dysregulation underlying epithelial–neuroendocrine transition–associated therapy resistance.

The Tip60-EP400 complex serves as a valuable downstream target for studying PRMT5's impact on splicing in MCC cells. This complex regulates chromatin remodeling, gene expression, DNA damage response, and cellular differentiation (Ikura et al, 2000; Fazzio et al, 2008; Chen et al, 2015; Jacquet et al, 2016). EP400 is particularly crucial in tumors with compromised SWI/SNF activity, indicating a compensatory interaction between chromatin-modifying complexes (Martin et al, 2023). MYC family proteins exploit the Tip60-EP400 complex to drive transcriptional programs linked to cancer and stem-like properties (Kim et al, 2010). In MCC, MCPyV ST stabilizes the interaction between MYCL and the Tip60-EP400 complex, promoting oncogenic transcriptional reprogramming (Cheng et al, 2017; Park et al, 2020). Despite high-resolution structural studies on Tip60-EP400 function (Chen et al, 2024; Li et al, 2024; Yang et al, 2024), the role of alternative splicing in its assembly and activity remains poorly understood. PRMT5 contributes by preventing exon skipping in *EP400* and *KAT5* while also methylating RUVBL1, a critical component of the complex (Koh et al, 2015; Clarke et al, 2017; Hamard et al, 2018). Understanding the interplay between PRMT5 and the Tip60-EP400 complex could uncover new insights into the connection between splicing dysregulation and chromatin remodeling.

Upon PRMT5 deficiency, most splicing events remain largely unchanged, with only a subset significantly affected (Braun et al, 2017). PRMT5's canonical role in RNA splicing through Sm protein methylation suggests that the selective nature of splicing deregulation upon PRMT5 deficiency is linked predominantly to weak 5' splice sites. Supporting this idea, conditional *Prmt5* deletion in neural stem/progenitor cells specifically caused intron retention and cassette-exon skipping primarily at transcripts with weak donor splice sites, reinforcing PRMT5's role as a critical regulator of both constitutive and alternative splicing (Bezzi et al, 2013).

However, this model fails to explain why many introns and cassette exons with weak 5' splice sites remain unaffected by PRMT5 loss, indicating the involvement of alternative mechanisms. Several PRMT family members, including PRMT1 and PRMT5, modulate alternative splicing by methylating serine/arginine-rich splicing factors (SRSFs), which can promote exon inclusion and intron removal by binding to exonic/intronic splicing enhancer sequences near splice sites and by antagonizing heterogeneous nuclear ribonucleoprotein (hnRNP) family proteins that bind to splicing silencer sequences (Kornblihtt et al, 2013). For instance, PRMT1-mediated asymmetric demethylation (ADMA) of SRSF1 has been linked to oncogenic exon inclusion in breast cancer (Li et al, 2023). Conversely, PRMT5-catalyzed SDMA of SRSF1 was shown to control cassette-exon choice in acute myeloid leukemia (Radzisheuskaya et al, 2019), underscoring the conserved role of SRSF1 methylation across cancers. In addition, N6-methyladenosine (m6A) modifications are a common internal RNA modification that also regulates alternative splicing by recruiting SRSF proteins to their binding motifs near splice junctions, though whether PRMT5 influences this recruitment remains unexplored (Xiao et al, 2016).

Here, we investigate PRMT5's role in MCC, focusing on its impact on splicing regulation by examining its influence on the Tip60-EP400 complex. Using a combination of full-length isoform sequencing (PacBio Iso-Seq) and short-read sequencing (Illumina RNA-seq), we demonstrate that PRMT5 inhibition prevents SRSF1 from binding m6A reader YTHDC1, leading to aberrant splicing. Our findings indicate that PRMT5-dependent SRSF1 modification is essential for full-length *KAT5* splicing, which is necessary for maintaining Tip60-EP400 activity. This study provides novel insights into PRMT5's essential role in RNA splicing, highlighting its broader significance in cellular homeostasis, gene expression control, and its connection to chromatin remodeling. These findings may also have implications for cancers exhibiting neuroendocrine differentiation or driven by MYC family proteins.

# Results

## Potent and specific PRMT5 inhibition in MCC cells

To investigate the role of PRMT5 in MCC, we first assessed the sensitivity of the established MCC cell line MKL-1 to PRMT5 inhibitors. PRMT5 inhibition exerts its effects through epigenetic deregulation, which may take 8–12 d to manifest (Kryukov et al, 2016; Mavrakis et al, 2016). This prolonged response is also consistent with the absence of a known corresponding arginine demethylase. To assess cell viability, we used a luminescent assay (CellTiter-Glo) to measure ATP levels, an indicator of metabolically active (live) cells. MKL-1 cells were treated with serially diluted PRMT5 inhibitors over multiple time points to determine their impact on viability. Consistent with findings in other cancer cell lines (Kryukov et al, 2016; Mavrakis et al, 2016), we observed that the PRMT5 inhibitor JNJ-64619178 (onametostat) effectively reduced MKL-1 cell viability at sub-nanomolar doses after 8 and 11 d of treatment. In contrast, a shorter 5-d

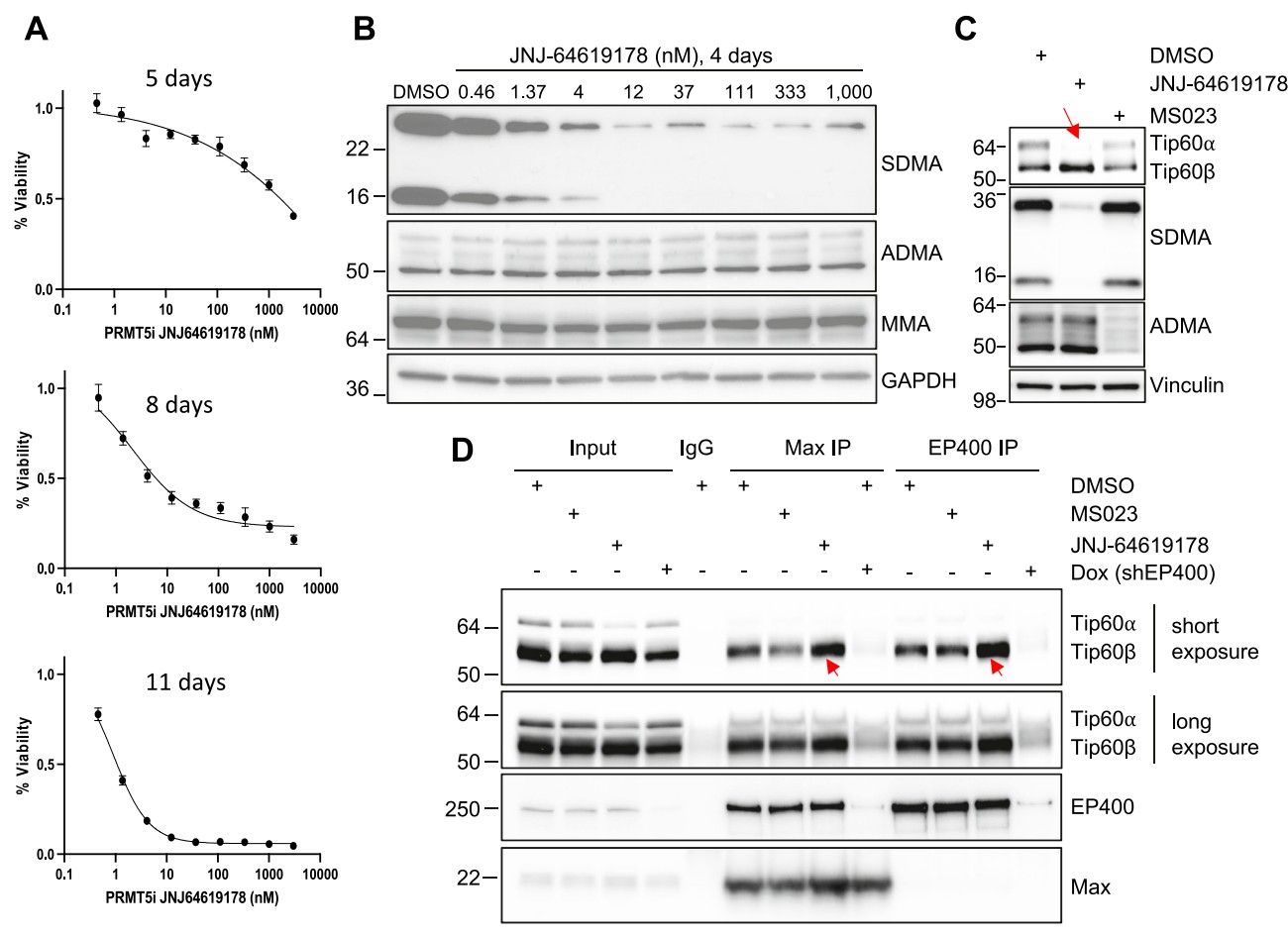

**Figure 1. PRMT5 inhibition leads to *KAT5* exon skipping and altered Tip60-EP400 complex composition.**
**(A)** Cell viability assay (CellTiter-Glo) of MKL-1 cells treated with the PRMT5 inhibitor JNJ-64619178 for 5, 8, and 11 d at indicated doses. **(B)** MKL-1 cells were treated for 4 d with serially diluted concentrations of JNJ-64619178, followed by immunoblotting with the indicated antibodies. **(C)** MKL-1 cells were treated with DMSO, 12 nM PRMT5 inhibitor (JNJ-64619178), and 111 nM type I PRMT inhibitor (MS023) for 3 d, followed by immunoblotting with the indicated antibodies. The arrow indicates a reduced full-length Tip60α protein level. **(D)** MKL-1 cell line with Dox-inducible EP400 shRNA was treated with DMSO, 111 nM type I PRMT inhibitor (MS023), 12 nM PRMT5 inhibitor (JNJ-64619178), or Dox for 3 d. Whole-cell lysates were immunoprecipitated with MAX or EP400 antibodies and immunoblotted with indicated antibodies. Arrows indicate the increased incorporation of the catalytically impaired Tip60β isoform protein into the ST-MYCL-EP400 complex upon PRMT5 inhibition. Both short and long exposures of the blot are shown to visualize abundant Tip60β and low-abundance Tip60α isoforms. Under these immunoprecipitation conditions, the low basal level of Tip60α remained near detection limits.
Source data are available for this figure.

treatment had minimal impact (Fig 1A). These results prompted us to evaluate whether JNJ-64619178 specifically inhibited PRMT5 catalytic activity at earlier time points. Western blot analysis revealed that 12 nM JNJ-64619178 effectively reduced SDMA levels while having no effect on ADMA or mono-methylarginine (MMA) levels (Fig 1B; see Supplemental Data 1 for uncropped and replicate blots). To confirm that this concentration did not induce significant cell death, we performed live/dead cell staining using calcein AM (live cells) and EthD-1 (dead cells). Fluorescence microscopy confirmed that after 5 d of treatment with 12 nM JNJ-64619178, most of the MKL-1 cells remained viable (Fig S1A).

To determine the earliest time point at which JNJ-64619178 effectively inhibits PRMT5, MKL-1 cells were treated with 12, 111, and 1,000 nM of the inhibitor for 1–4 d. PRMT5 inhibition was first detectable after 3–4 d, as assessed by SDMA Western blotting.

Increasing the inhibitor concentration did not significantly accelerate this process (Fig S1B).

Similarly, we tested another potent and selective PRMT5 inhibitor, LLY-283. Western blot analysis from a time-course experiment showed that treatment with 37–111 nM LLY-283 for 4 d markedly reduced SDMA levels without affecting ADMA or MMA (Fig S1C). CellTiter-Glo viability assays further revealed that the half-maximal effective concentration (EC50) of LLY-283 after 8 d of treatment was ~10 nM, which was higher than the EC50 of JNJ-64619178 under the same conditions (Fig S1D).

To ensure the observed effects were specifically due to PRMT5 inhibition rather than cytotoxicity, we established experimental conditions treating MKL-1 cells with 12 nM JNJ-64619178 for ≤5 d. Under these optimal conditions, changes detected are attributed to specific PRMT5 inhibition. All subsequent experiments used these conditions unless otherwise

indicated; lower concentrations were defined as suboptimal conditions.

## Inhibition of type I PRMTs does not reduce MCC cell viability as effectively as PRMT5 inhibition

To compare the impact of PRMT5 inhibition with inhibition of type I PRMTs on cell viability, we tested MS023, a broad-spectrum inhibitor targeting type I PRMTs (PRMT1, PRMT3, PRMT4, PRMT6, PRMT8) without affecting PRMT5, a type II PRMT. CellTiter-Glo viability assays showed that MS023 exhibited a higher EC50 compared with PRMT5 inhibitors, indicating reduced efficacy in decreasing cell viability (Fig S1E).

Western blot analysis of ADMA levels further corroborated these findings, showing a significant reduction at 37–111 nM MS023 (Fig S1F). Although MS023 did not alter SDMA levels, it led to an increase in MMA levels upon ADMA reduction, suggesting a compensatory mechanism. These results support the notion that type I PRMTs are the primary mediators of protein methylation in mammalian cells, whereas PRMT5-mediated SDMA modifications represent a smaller fraction of total protein methylation (Pawlak et al, 2000; Tang et al, 2000). Consequently, PRMT5 inhibition does not induce an increase in MMA levels (Figs 1B and S1C).

## PRMT5 enhances Tip60-EP400 histone acetyltransferase activity by promoting full-length Tip60 expression

Next, we investigated whether PRMT5 enhances the activity of the Tip60-EP400 complex in MCC cells. In an MKL-1 stable cell line expressing doxycycline (Dox)-inducible EP400 shRNA, treatment with either Dox or 12 nM JNJ-64619178 resulted in reduced acetylation of histone H4 at lysine 5, 12, and 16, as well as decreased acetylation of the H2A.Z histone variant (Fig S1G and H). Notably, the combination of Dox and JNJ-64619178 further enhanced this reduction in histone acetylation (Fig S1G and H). Because the Tip60-EP400 complex, also known as nucleosome acetyltransferase of H4 (NuA4), is a major histone acetyltransferase complex, these findings align with previous reports demonstrating that PRMT5 enhances Tip60-EP400 activity in other cell types (Clarke et al, 2017; Hamard et al, 2018).

Because PRMT5 is required to maintain proper splicing of full-length *EP400* in a *MYC*-driven lymphomagenesis mouse model and has also been reported to promote full-length *KAT5* splicing in hematopoietic cells (Koh et al, 2015; Hamard et al, 2018), we tested the effect of PRMT5 inhibition on their splicing in MCC cells using Western blot analysis. PRMT5 inhibition significantly reduced full-length Tip60 levels, an effect not observed with 111 nM MS023 treatment (Fig 1C and D) or in the context of Dox-induced EP400 knockdown (Fig 1D). However, full-length EP400 protein levels were not significantly altered by PRMT5 inhibition in MKL-1 cells (Fig 1D).

Given that the Tip60-EP400 complex forms a stable interaction with MYCL, MAX, and MCPyV ST antigen in MCC cells, which is an essential driver of MCC tumorigenesis, we next examined whether PRMT5 inhibition affects the composition of the ST-MYCL-EP400 complex. Immunoprecipitation of the ST-MYCL-EP400 complex components using MAX or EP400 antibodies revealed increased incorporation of a spliced Tip60 isoform (Tip60β) (Hamard et al, 2018), which is presumed to be catalytically impaired, upon PRMT5 inhibition (Fig 1D). These findings suggest that PRMT5 regulates the composition of the ST-MYCL-EP400 complex through alternative splicing, thereby altering Tip60-EP400 complex activity.

## PRMT5 represses p53 and MYC targets, as suggested by global gene expression patterns

Based on these assays, we conducted sequencing-based gene expression analyses on MKL-1 cells treated with 12 nM JNJ-64619178 for 4 d to further elucidate the role of PRMT5 in splicing regulation and its potential contribution to MCC tumorigenesis. We generated paired Illumina short-read and PacBio long-read RNA-seq datasets from four DMSO-treated and four JNJ-64619178–treated MCC samples (Table S1).

Differential expression analysis revealed that 2,616 genes were up-regulated (LFC > 0) in PRMT5 inhibitor (JNJ-64619178)–treated samples and 2,793 genes were down-regulated (LFC < 0) at an adjusted *P*-value < 0.05. Detailed differential expression results can be found in Table S2. An MA plot showing expression changes relative to mean expression levels and a volcano plot visualizing the magnitude and significance of differential expression are shown in Fig 2A and B, respectively.

Gene set enrichment analysis (GSEA) was performed to identify biological pathways and gene sets significantly enriched in the dataset. A total of 50 curated gene sets from the hallmark gene sets of the Molecular Signatures Database (MSigDB) were included in the analysis. These hallmark gene sets represent well-defined biological processes and are derived from systematic clustering of gene expression data, identifying coherent biological states or processes that are more robust and reproducible compared with other gene sets. Upon PRMT5 inhibition, 22 of the 50 hallmark gene sets were up-regulated, with 16 of these reaching statistical significance at an FDR < 0.25 (Table S3). Seven gene sets were significantly enriched with nominal *P*-value < 0.01, and 13 gene sets were significantly enriched with nominal *P*-value < 0.05 (Fig 2C). The "Oxidative Phosphorylation" pathway was significantly enriched, suggesting alterations in cellular energy metabolism associated with PRMT5 inhibition. Other notable enriched pathways include "TNFα Signaling via NF-kB" and "UV Response Up," which were significant at a nominal *P*-value < 0.05. These results highlight key biological processes related to stress responses, immune signaling, and cellular metabolic changes. The full list of up-regulated gene sets also includes pathways related to "DNA repair," "mTORC1 signaling," and "inflammation," offering further insight into the broader effects of PRMT5 inhibition (Table S3). Among the significantly enriched pathways, the most up-regulated were "MYC Targets" (v1 and v2) and the "P53 Pathway," both exhibiting strong enrichment with nominal *P*-values near zero (Fig 2D); the functional implications of MYC and P53 activation are addressed below in the Discussion section.

To conduct hypothesis-driven GSEA, we used a curated set of gene sets from lomics.ai, specifically chosen to align with biological processes relevant to PRMT5 inhibition. This targeted approach allowed for a more focused analysis compared with

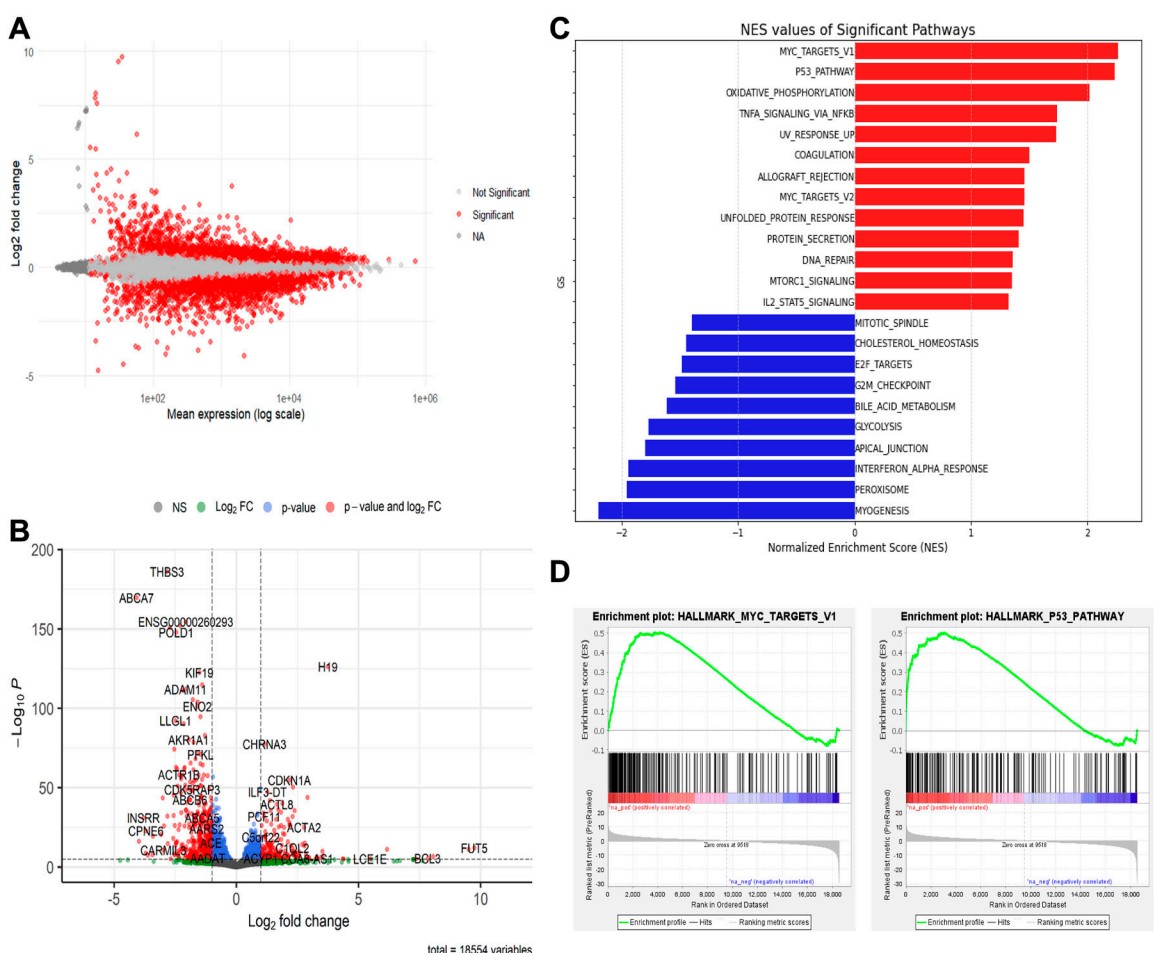

**Figure 2. PRMT5 inhibition leads to the up-regulation of MYC and p53 pathways.**
**(A)** MA plot of differential gene expression results. The x-axis represents the mean expression levels of genes, whereas the y-axis shows the $\log_2$ fold changes (LFC) between conditions. Genes that are significantly differentially expressed (adjusted *P*-value < 0.05) are highlighted in red. **(B)** Volcano plot of differential gene expression results. The x-axis shows LFC, and the y-axis shows $-\log_{10}$ (adjusted *P*-value). Genes with |LFC| > 1 and *P* < 0.05 are in red; those not meeting *P*-value or LFC thresholds are in blue and green, respectively; others are in gray. Gene labels were automatically selected by the EnhancedVolcano R package, which prioritizes the most statistically significant and highly differentially expressed genes and places as many labels as possible without overlapping. **(C)** Normalized enrichment score (NES) values for significantly up- and down-regulated pathways from the gene set enrichment analysis using the MSigDB hallmark gene sets. Pathways up-regulated in PRMT5-inhibited samples are shown in red, whereas down-regulated pathways are shown in blue. **(D)** Gene set enrichment analysis enrichment plots for the top two up-regulated pathways: MYC Targets V1 and P53 Pathway. Each plot displays the ranked gene list scores and enrichment score (ES) curve. Ranked gene lists were obtained from differential gene expression results and ordered by the Wald statistics, which is the ratio of the logFC to standard error.

traditional GSEA using the MSigDB, as it emphasized pathways directly associated with PRMT5's regulatory role. The LLM selected 40 gene sets encompassing a wide range of biological processes, particularly those related to gene expression regulation, chromatin modifications, and cellular maintenance. These include processes such as histone modification, RNA splicing, alternative splicing, transcriptional regulation, epigenetic modifications, and cell cycle regulation. The sets also focus on pathways involved in cancer development, stem cell maintenance, immune response, apoptosis, and transcriptional elongation, among others, reflecting a broad scope of cellular functions and regulatory mechanisms. The analysis identified significant enrichment in 38 of the 40 curated gene sets in PRMT5-inhibited samples, with 30 gene sets significantly enriched at FDR < 0.25. Among these, 11 gene sets reached a nominal *P*-value < 0.01, and 19 gene sets were significant

at < 0.05. In contrast, only two gene sets were down-regulated, and none met significance criteria at any threshold. A full list of the gene sets included in the analysis and GSEA results can be seen in Table S4. These results indicate a strong link between PRMT5 activity and the curated pathways, further supporting the diverse cellular functions regulated by PRMT5.

## PacBio full-length isoform differential expression and functional enrichment

A total of 6,296 isoforms were identified as differentially expressed (*P*-value < 0.05) in PRMT5-inhibited samples: 2,942 up-regulated and 3,354 down-regulated (Fig 3A; Table S5). Functional enrichment analysis of differentially expressed isoforms was conducted using gProfiler (Kolberg et al, 2023), categorizing results into GO

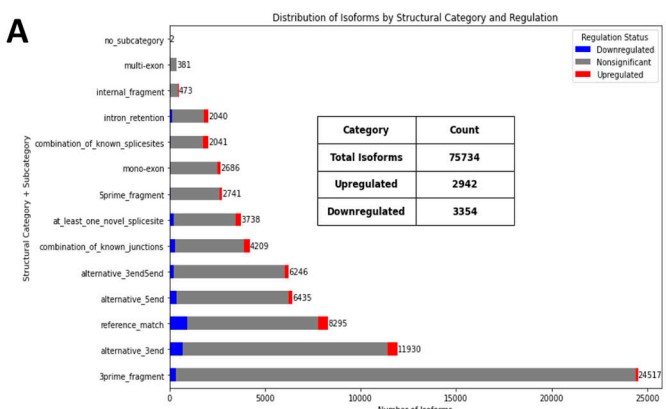

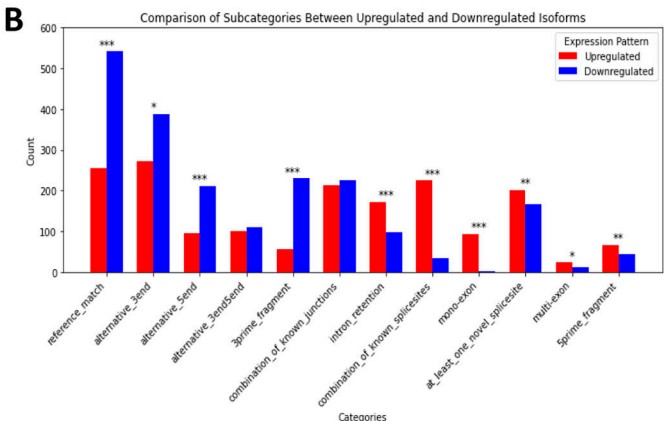

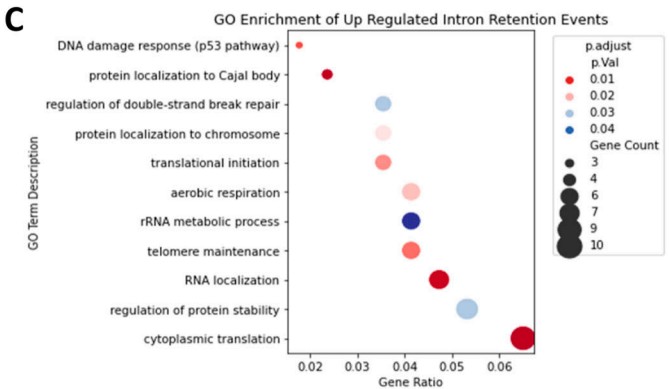

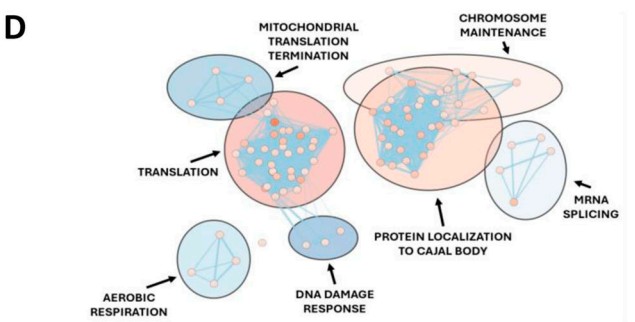

Biological Process (GO BP), KEGG pathways, and REACTOME pathways. In total, we identified 633 significantly enriched pathways among up-regulated isoforms and 500 among down-regulated isoforms (FDR < 0.05, $P$ < 0.05) (Table S6). Notably, double-strand break repair was significantly enriched, with six Tip60-EP400 complex subunits—*YEATS4*, *MEAF6*, *DMAP1*, *ING3*, *MORF4L2*, and *KAT5*—showing differential expression. This highlights a key regulatory role of PRMT5 in double-strand break repair via the Tip60-EP400 complex in MCC, supporting the concept of synergy between PRMT5 inhibitors and DNA-damaging agents in cancer cells (Gillespie et al, 2024).

### Full-length isoform (PacBio Iso-Seq) gene expression levels correlate with those of short-read (Illumina RNA-seq) datasets

To compare the expression patterns of the full-length isoform and short-read datasets, we investigated the differentially expressed gene (DEG) sets from gene-level aggregated data. In the full-length isoform dataset, 1,327 up-regulated genes and 1,562 down-regulated genes were identified (Table S7). We found that 948 genes were commonly up-regulated, and 1,038 genes were commonly down-regulated in both datasets (Fig S2A). In contrast, 1,663 genes were uniquely up-regulated in short-read data, whereas 379 genes were uniquely up-regulated in the full-length isoform dataset. Similarly, 1,745 genes were uniquely down-regulated in the short-read dataset, and 524 genes were uniquely down-regulated in the full-length isoform dataset (Fig S2B). The correlation analysis on normalized counts showed high positive correlations between the two platforms for both the control condition (Pearson's correlation: 0.711, Spearman's correlation: 0.703) and treated condition (Pearson's correlation: 0.671, Spearman's correlation: 0.653) (Fig S2C and D). This suggests that despite the differences in sequencing technologies, the overall expression patterns in the control and treated conditions were consistent between Illumina short-read and PacBio full-length isoform sequencing. The significant overlap of DEGs between the platforms suggests a shared biological response to treatment, whereas the unique gene sets identified in each platform indicate possible platform-specific sensitivities or biases in detecting gene expression changes.

To further investigate transcript-level regulation, we compared the Illumina DE genes (Table S2) with genes containing at least one differentially expressed isoform in the PacBio dataset (Table S5).

This analysis identified 3,092 genes with differential isoform expression, of which 1,735 overlapped with the 5,394 DE genes from Illumina. Notably, 3,659 genes were uniquely identified as DE in the Illumina dataset, whereas 1,357 were uniquely identified based on isoform-level changes in PacBio. These findings emphasize the complementary nature of gene-level and isoform-level analyses, suggesting that long-read sequencing reveals regulatory variation at the isoform level that may be missed by traditional short-read approaches.

### hnRNP genes exhibit extensive isoform diversity in MCC cells

The expression of most of the isoforms was not significantly changed by PRMT5 inhibition (Fig 3A), supporting the selective nature of splicing deregulation in PRMT5-deficient cells. The mean number of isoforms per gene was 7.32 (median: 4.0), with a maximum of 146 isoforms (Fig S3A). Among the top 50 genes with the highest isoform abundance in our dataset, many encode RNA-binding proteins (RBPs) that contribute to splice-site selection, alternative splicing, and spliceosome function, including several heterogeneous nuclear ribonucleoprotein (hnRNP) family members (Fig S3A).

We detected a total of 865 isoforms across 29 annotated hnRNP genes, with isoform counts per gene ranging from 1 to 146. Of these, 80 isoforms from 11 unique hnRNP genes were significantly differentially expressed in PRMT5-deficient cells (adjusted $P$ < 0.05; Table S5). The most pronounced isoform-level changes were seen in HNRNPK and HNRNPC, with 33 and 24 differentially expressed isoforms, respectively. At the gene level, three hnRNP genes (*HNRNPUL2-BSCL2*, *HNRNPL*, and *HNRNPH1*) showed significant expression changes (Table S2). Functionally, hnRNPs bind intronic/exonic splicing silencers to inhibit exon inclusion, whereas SRSF proteins bind enhancer elements to promote inclusion, thereby antagonizing hnRNP activity (Kornblihtt et al, 2013).

### PRMT5 inhibition leads to increased intron retention and exon skipping events

The relationship between the total isoforms for a given gene and weighted gene expression change (average $\log_2$FoldChange) is illustrated (Fig S3B). Genes with fewer isoforms exhibit larger expression changes, suggesting that PRMT5 inhibition has a stronger effect on genes with limited isoform diversity. In contrast,

**Figure 3. PRMT5 inhibition leads to increased intron retention and exon skipping events.**
**(A)** Distribution of isoforms across different structural categories and their regulation status. Isoforms significantly up-regulated in PRMT5-inhibited samples ($P$ < 0.05) are shown in red; significantly down-regulated isoforms are shown in blue. The x-axis represents the number of isoforms, and the y-axis indicates structural subcategories as defined by the SQANTI classification system. Reference_match, alternative_5end, alternative_3end, and alternative_3end5end correspond to isoforms with known exon structures that match the reference transcript model. 5prime_fragment, 3primefragment, mono-exon, and intron_retention represent isoforms that lack terminal exons but still align to known splice junctions. Combination_of_known_junctions and combination_of_known_splicesites refer to isoforms that use only annotated splice sites in new combinations. At_least_one_novel_splicesite and multi-exon represent isoforms that include at least one previously unannotated splice site. **(B)** Comparison of up-regulated (red) and down-regulated (blue) isoforms across various structural categories. Statistical significance is annotated on the bars using asterisks, based on $P$-values from a chi-squared test. Asterisks indicate statistical significance: $P$ < 0.05 (*), $P$ < 0.01 (**), and $P$ < 0.001 (***). **(C)** Dot plot visualizing Gene Ontology (GO) enrichment analysis for isoforms with increased intron retention. The x-axis represents the gene ratio (the proportion of input genes associated with each GO term). The y-axis lists the enriched GO-term descriptions. Dot size corresponds to the number of genes associated with each term, whereas the dot color represents the adjusted $P$-value, with the red-blue gradient indicating significance levels. Terms are sorted by gene ratio for clarity. **(D)** Cytoscape Enrichment Map of significantly enriched gene sets in all up-regulated intron retention containing isoforms. Gene sets are colored based on $-\log_{10}$ ($P$-value) with darker red indicating lower $P$-values. Clusters were completed using AutoAnnotate and are labeled based on the pathway in each cluster with the lowest adjusted $P$-value.

genes with many isoforms show more stable overall expression. To determine whether PRMT5 inhibition alters alternative splicing patterns, we examined the distribution of significantly up-regulated and down-regulated isoforms across structural subcategories (Fig 3B). A chi-squared test revealed significant differences in the regulation of specific isoform types. Intron retention (IR), exon skipping (ES; also known as "combination_of_known_splice_sites"), mono-exons, novel splice sites, and 5′ fragments were more abundant in significantly up-regulated isoforms in PRMT5 inhibitor–treated cells. Exact reference matches, alternative 3′ end, alternative 5′ ends, and 3′ fragments were found to be more abundant in significantly down-regulated isoforms (Fig 3B). These findings indicate that PRMT5 inhibition disrupts normal splicing, promoting the production of alternative and often incomplete transcript isoforms.

To further explore the biological significance of differentially expressed isoform structural categories, we analyzed IR and ES events separately (Table S8). Among up-regulated IR events, we identified 39 significantly enriched GO BP terms and 49 REACTOME pathways, with translation emerging as a major affected process (Fig 3C). Specifically, GO:0022613 ribonucleoprotein complex biogenesis, GO:0002181 cytoplasmic translation, and GO:0008380 RNA splicing were significantly enriched, highlighting the role of PRMT5 in RNA metabolism. In addition, GO:0009060 aerobic respiration was notable, suggesting that PRMT5 regulates mitochondrial oxidative metabolism. Within these pathways, *EIF4E*, a key translation initiation factor, was significantly up-regulated, consistent with previous findings (Braun et al, 2017). Furthermore, GO:1904867 protein localization to the Cajal body and GO:0090670 RNA localization to the Cajal body were enriched, suggesting that PRMT5 loss may disrupt snRNP complex biogenesis and Cajal body function, supporting the model proposed by Braun et al (2017). In contrast, down-regulated IR events were associated with six significantly enriched GO BP pathways, primarily related to protein ubiquitination and antigen presentation, including HLA-E regulation. For ES events, no significantly enriched down-regulated pathways were identified; however, up-regulated ES events were enriched in 21 GO BP terms, 8 KEGG pathways, and 37 REACTOME pathways. Notably, "DNA repair" and "aerobic respiration and respiratory electron transport" were among the top enriched pathways, further linking PRMT5 inhibition to genome stability and mitochondrial function. In addition, "Antigen processing—Cross presentation" was enriched, with HLA-C and HLA-E identified as key DEGs, indicating that PRMT5 regulates antigen presentation. The network analysis revealed key clusters of genes involved in protein localization, translation, mRNA splicing, aerobic respiration, and DNA damage response (Fig 3D). This highlights how these splicing changes might disrupt cellular function and suggests potential regulatory relationships between splicing factors and other pathways.

## PRMT5 regulates the ST-MYCL-EP400 complex composition through splicing

PRMT5 inhibition significantly alters the expression of several subunits in the ST-MYCL-EP400 complex, including *KAT5* (Tip60), *EP400*, *RUVBL1*, *MORF4L2*, and *MYCL* (Tables 1 and S2). Analysis of

isoform diversity using full-length isoform sequencing revealed extensive alternative splicing events. The number of isoforms per gene varied considerably, with ACTB, MORF4L1, and MEAF6 showing the highest diversity, with over 30 isoforms each (Table 1). Several splicing categories were detected, including reference matches, alternative 3′ and 5′ end usage, intron retention, and novel splice sites. Notably, KAT5 exhibited multiple splicing events, including alternative 3′ and 5′ end usage, reference matches, and 3′ prime fragments. Among differentially expressed isoforms, significant up-regulation was observed in *BRD8*, *EPC1*, *VPS72*, *ACTB*, *MEAF6*, and *MYCL* in PRMT5 inhibitor–treated cells, whereas significant down-regulation was detected in *KAT5*, *DMAP1*, *MORF4L2*, and *YEATS4*. Several genes, such as *TRRAP* and *RUVBL2*, displayed multiple isoforms without significant changes in expression (Table S5). Overall, these findings highlight the extensive isoform diversity of the Tip60-EP400 complex. Instead of forming a single, consistent complex, it may exist as a heterogeneous pool of related variants. This also suggests a regulatory role of PRMT5 in shaping the composition of the complex (Table 1).

## PRMT5 inhibition induces *KAT5* exon 5 skipping

Given that the double-strand break repair pathway, including Tip60-EP400 complex subunits, was identified through functional enrichment analysis of our full-length isoform sequencing data, we focused on *KAT5* to further explore PRMT5-mediated splicing regulation. We identified two *KAT5* isoforms (PB.15416.14 and PB.15416.41) that were significantly down-regulated in PRMT5-inhibited samples. These isoforms have alternative 5′ ends but retain exon 5 (position 65713348–65713503) and lack an optional intron between exon 1 and exon 2 (Tables 1 and S5). This is consistent with our observation of decreased full-length Tip60 protein levels upon PRMT5 inhibition, as shown by Western blotting (Fig 1C and D).

To further validate the skipping of *KAT5* exon 5 upon PRMT5 inhibition, we used the replicate Multivariate Analysis of Transcript Splicing (rMATS) pipeline to analyze short-read sequencing data and identify alternatively spliced exons (Shen et al, 2014). Gene Ontology (GO) analysis of these genes revealed significant enrichment in mRNA binding and DNA repair pathways, including *KAT5*, consistent with Tip60's well-established role in DNA damage repair (Fig 4A) and our full-length isoform data. Consistent with our long-read sequencing analysis, short-read sequencing confirmed significant skipping of *KAT5* exon 5 in JNJ-64619178–treated cells (Fig 4B and C). The larger (≈64 kD) and smaller (≈55 kD) bands we detected align with previous findings in PRMT5-knockout hematopoietic stem and progenitor cells (HSPCs), corresponding to the Tip60α and exon 5–skipped, catalytically impaired Tip60β isoforms. Although exon 5 does not encode residues within the catalytic domain, its skipping removes a proline-rich regulatory region critical for efficient autoacetylation and full histone acetyltransferase (HAT) activity (Ran & Pereira-Smith, 2000; Hamard et al, 2018). A key advantage of the rMATS pipeline is its ability to predict RBPs based on enriched binding motifs near alternatively spliced sites. Potential RBPs involved in exon inclusion and intron removal can be identified by uploading rMATS output files to the RNA Map Analysis and Plotting Server 2

**Table 1. Genes of interest from Tip60-EP400 protein complex with a summary of differential expression results and isoform categories.**

| Gene | log$_2$FoldChange | Padj | Isoforms | Significant_up | Significant_down | Summary_category |
|---|---|---|---|---|---|---|
| KAT5* | −0.41076 | 0.00022547 | 13 | 0 | 2 | Truncated transcript\|Canonical isoform\|Alternative splicing\|Other |
| EP400* | −0.28062 | 0.03810907 | 1 | 0 | 0 | Other |
| TRRAP | −0.13833 | 0.41436348 | 4 | 0 | 0 | Other |
| BRD8 | 0.15795 | 0.35760197 | 15 | 1 | 0 | Truncated transcript\|Canonical isoform\|Other |
| EPC1 | 0.16909 | 0.13996817 | 6 | 1 | 0 | Truncated transcript\|Canonical isoform\|Alternative splicing |
| EPC2 | 0.06437 | 0.70123456 | 4 | 0 | 0 | Truncated transcript |
| MBTD1 | −0.18101 | 0.07057304 | 2 | 0 | 0 | Truncated transcript\|Novel isoform |
| DMAP1 | −0.14545 | 0.27985282 | 13 | 0 | 2 | Alternative splicing\|Truncated transcript\|Canonical isoform\|Other\|Novel isoform |
| ING3 | 0.02489 | 0.92318757 | 8 | 1 | 1 | Canonical isoform\|Alternative splicing |
| VPS72 | −0.10319 | 0.65086163 | 13 | 2 | 1 | Truncated transcript\|Other\|Alternative splicing\|Novel isoform |
| RUVBL2 | 0.11101 | 0.36815264 | 15 | 0 | 0 | Alternative splicing\|Truncated transcript\|Canonical isoform\|Other\|Novel isoform |
| RUVBL1* | 0.40189 | 0.00028326 | 14 | 0 | 0 | Alternative splicing\|Truncated transcript\|Canonical isoform\|Other\|Novel isoform |
| ACTL6A | 0.13178 | 0.23843247 | 13 | 0 | 0 | Truncated transcript\|Canonical isoform\|Alternative splicing\|Other |
| ACTB | 0.15825 | 0.06197625 | 33 | 1 | 0 | Truncated transcript\|Canonical isoform\|Alternative splicing\|Novel isoform |
| MORF4L1 | 0.23859 | 0.08291186 | 33 | 0 | 0 | Truncated transcript\|Alternative splicing\|Other\|Novel isoform |
| MORF4L2* | −0.59104 | $1.92 \times 10^{-11}$ | 18 | 2 | 4 | Alternative splicing\|Truncated transcript\|Canonical isoform\|Other\|Novel isoform |
| YEATS4 | −0.04442 | 0.78374579 | 7 | 1 | 2 | Truncated transcript\|Alternative splicing\|Other\|Novel isoform |
| MRGBP | 0.17219 | 0.41749438 | 3 | 0 | 0 | Truncated transcript\|Alternative splicing\|Other |
| MEAF6 | 0.11359 | 0.36804976 | 31 | 2 | 7 | Truncated transcript\|Canonical isoform\|Alternative splicing\|Other |
| H2AZ1 | 0.04005 | 0.85645869 | 10 | 0 | 0 | |
| MYCL* | 0.37867 | $3.24 \times 10^{-9}$ | 15 | 0 | 0 | Canonical isoform\|Other\|Alternative splicing\|Novel isoform |

Log$_2$FoldChange and adjusted *P*-values are from differential expression analysis of Illumina short-read data. The number of isoforms was obtained using PacBio full-length isoform sequencing with structural categories based on the SQANTI isoform classification system. * Genes marked with an asterisk are subunits of the ST-MYCL-EP400 complex whose expression was significantly altered upon PRMT5 inhibition.

(rMAPS2) (Hwang et al, 2020). These RBPs may be targets of PRMT5-mediated methylation and could play a role in the selective regulation of alternative splicing by PRMT5.

### SRSF1-binding motif is enriched near alternatively spliced sites upon PRMT5 inhibition

Given that PRMT1-mediated ADMA modification of SRSF1 has been reported to promote oncogenic exon inclusion and drive breast tumorigenesis (Li et al, 2023), we investigated whether SRSF1 is involved in alternative splicing regulation by PRMT5 in MCC cells.

rMAPS2 identified 6,455 statistically significant exon-skipping events and 2,596 significant exon-inclusion events in PRMT5-inhibited cells, compared with 14,256 background events where exon usage was not significantly affected by PRMT5 inhibition. Analysis revealed that SRSF1-binding motifs were enriched near splice sites flanking skipped exons upon PRMT5 inhibition, whereas these motifs were less enriched near splice sites of exons with increased inclusion (Fig 4D). This suggests that PRMT5 can potentially methylate SRSF1 to promote exon inclusion, and PRMT5 inhibition shifts splicing dynamics toward exon skipping.

To further examine SRSF1's role in PRMT5-regulated splicing, we also used rMATS, rMAPS2, and short-read sequencing data to analyze intron retention events. Compared with 2,472 intron

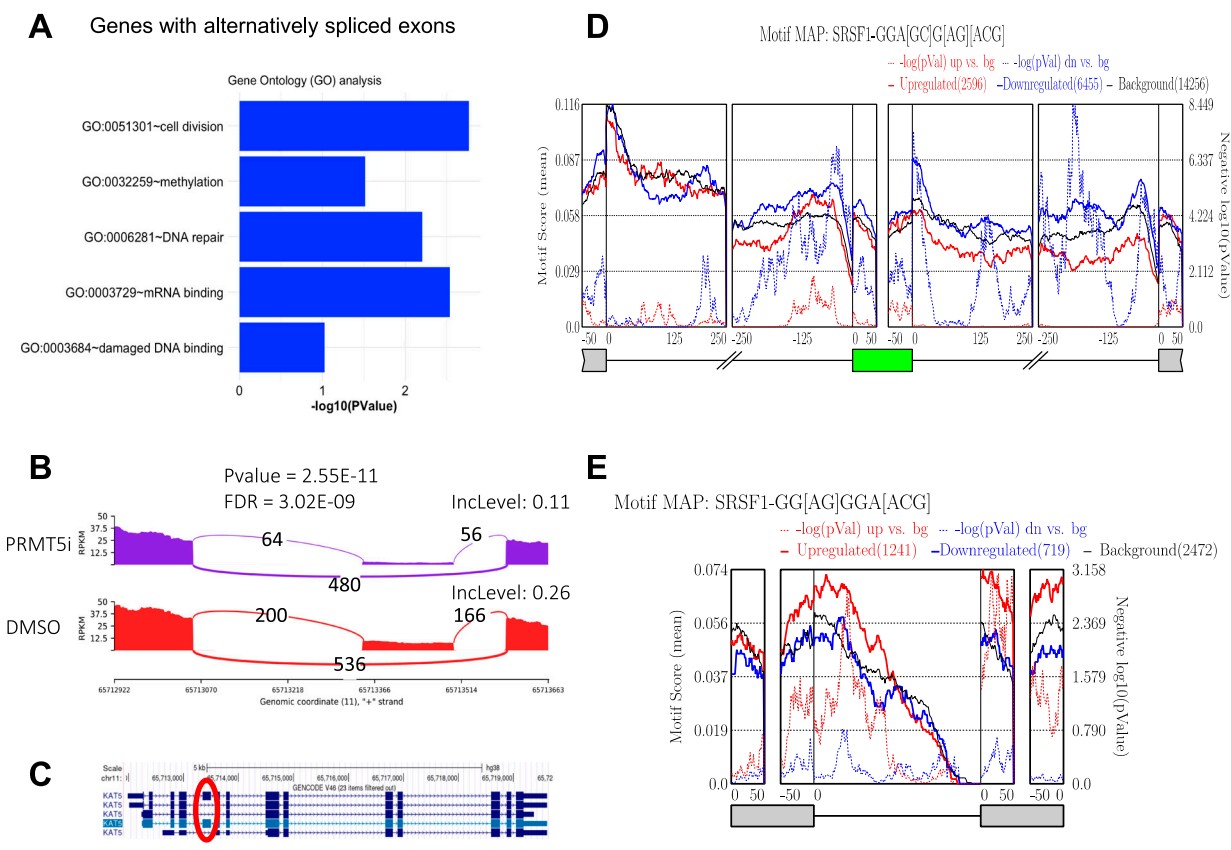

**Figure 4. Role of SRSF1 in promoting exon inclusion.**
**(A)** Alternative splicing of Illumina short-read data was analyzed using rMATS, and genes with alternatively spliced exons were subjected to Gene Ontology (GO) analysis using ClusterProfiler. **(B)** Sashimi plot depicting significant alternative splicing of *KAT5* exon 5 between PRMT5 inhibitor (JNJ-64619178) and DMSO-treated cells based on Illumina short-read RNA-seq. **(C)** Skipped *KAT5* exon in JNJ-64619178–treated cells, consistent with *KAT5* exon 5 annotation from the UCSC Genome Browser. **(D)** SRSF1-binding motifs cluster around cassette exons that are skipped after PRMT5 inhibition. Meta-plots were generated with rMAPS2 using a 50-nt sliding window. Solid traces (motif score, left y-axis): blue line, exons whose inclusion decreases after JNJ-64619178 treatment (down-regulated; n = 6,455); red line: exons whose inclusion increases after treatment (up-regulated; n = 2,596); gray line, cassette exons with unchanged inclusion (background; n = 14,256). The solid curves plot the mean SRSF1 motif score at each position. Dotted traces (right y-axis): corresponding –log$_{10}$ (*P*-values) from a Wilcoxon rank-sum test comparing motif scores in each regulated set with background. Prominent peaks in the blue profile, accompanied by high –log$_{10}$ (*P*-values), reveal significant enrichment of SRSF1-binding motifs both upstream and downstream of exons that become skipped when PRMT5 activity is lost, consistent with diminished SRSF1-dependent exon inclusion under PRMT5 inhibition. **(D, E)** rMAPS2 meta-plot of SRSF1-motif density flanking introns whose retention changes after PRMT5 inhibition; same axis format as (D).

retention events not significantly altered, PRMT5 inhibition caused a significant increase in retention in 1,241 introns and promoted the removal of 719 retained introns. Notably, SRSF1-binding motifs were enriched in retained introns upon PRMT5 inhibition, suggesting that PRMT5 also promotes intron removal through SRSF1-mediated splicing regulation (Fig 4E).

### SRSF1 overexpression rescues PRMT5 inhibition–induced *KAT5* exon skipping

To validate the role of SRSF1 in PRMT5-mediated splicing regulation, we examined its impact on *KAT5* splicing. Using an MKL-1 stable cell line with Dox-inducible SRSF1 expression, we found that SRSF1 overexpression produced a significant partial restoration of full-length Tip60α when cells were treated with a suboptimal dose (1.37 nM) of JNJ-64619178 (mean ± SD, n = 3; paired two-tailed *t* test, *P* < 0.05). At the optimal dose (12 nM), SRSF1 expression did not rescue *KAT5* exon skipping (*P* > 0.05)

(Fig 5A; see Supplemental Data 1 for uncropped and replicate blots). This is consistent with prior evidence that SDMA on SRSF1 is required for its splicing-activator function (Radzisheuskaya et al, 2019). These findings support our RNA-seq analysis, which identified SRSF1-binding motif enriched near splice sites of skipped exons upon PRMT5 inhibition. A potential model suggests that PRMT5 methylates arginine residues of SRSF1, facilitating its recruitment to exonic or intronic splicing enhancers near splice sites to promote exon inclusion (Kornblihtt et al, 2013). This regulatory mechanism may influence the splicing of key DNA damage response genes, such as *KAT5*.

### PRMT5 is not transactivated by the ST-MYCL-EP400 complex

MYC has been suggested to act as a general amplifier of RNA production, potentially overwhelming the splicing machinery in *MYC*-driven cancer cells because of excessive total RNA output (Lin et al, 2012; Nie et al, 2012). To counterbalance this, MYC also up-

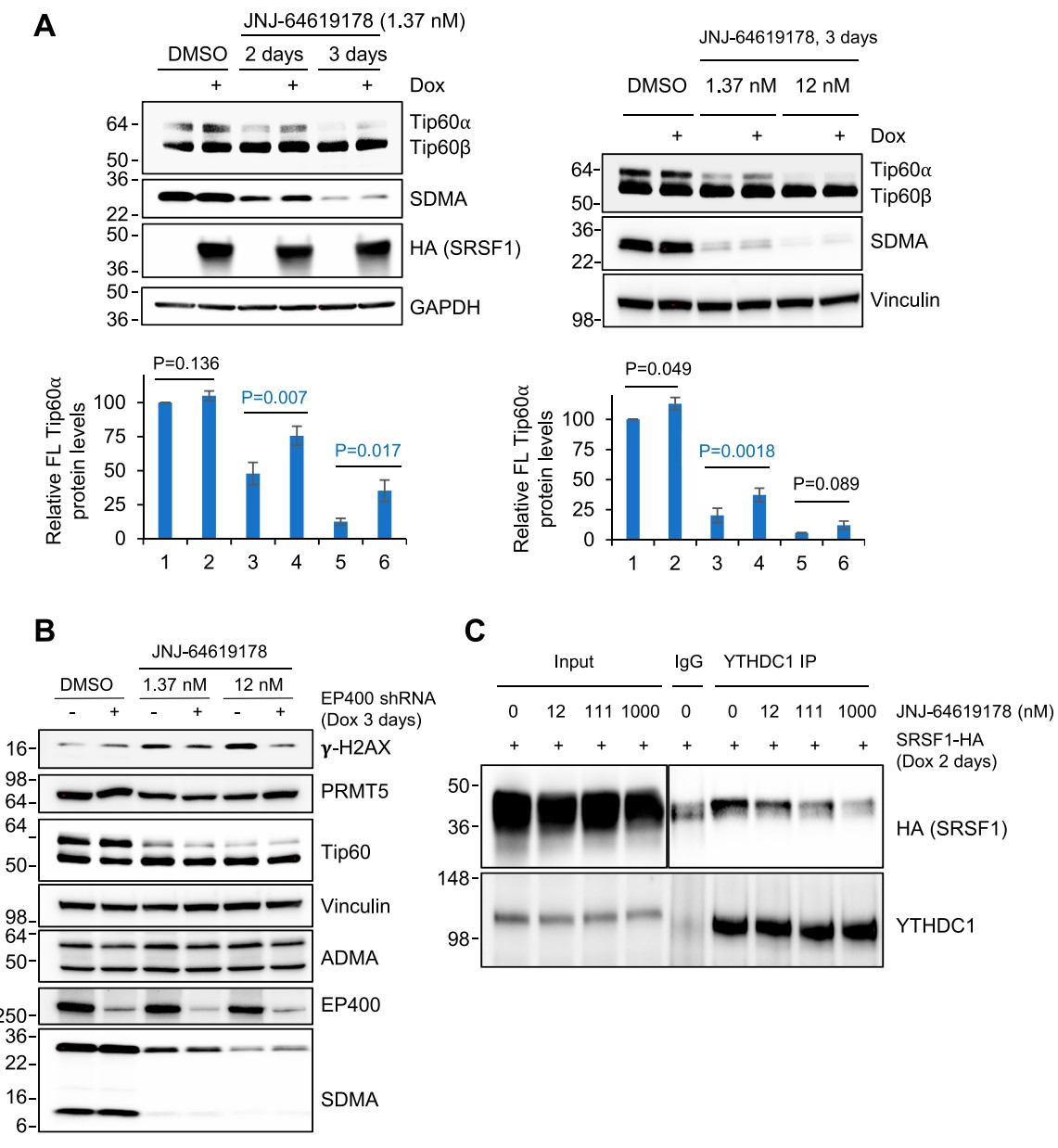

**Figure 5. SRSF1 contributes to PRMT5-mediated alternative splicing.**
**(A)** Western blot analysis of MKL-1 cells treated with suboptimal (1.37 nM) or optimal (12 nM) JNJ-64619178, showing rescue of Tip60α exon inclusion by doxycycline-induced HA-SRSF1 expression. Bar graphs summarize quantification of full-length Tip60α levels normalized to housekeeping proteins from three independent experiments (n = 3, mean ± SD). Statistical significance was determined by a paired two-tailed t test. **(B)** MKL-1 cells with Dox-inducible EP400 shRNA were treated with JNJ-64619178 and Dox, then analyzed by Western blotting using the indicated antibodies. **(C)** MKL-1 cells expressing Dox-inducible HA-tagged SRSF1 were treated with serially diluted JNJ-64619178 for 2 d. Whole-cell lysates were immunoprecipitated with a YTHDC1 antibody and analyzed by Western blot using HA (SRSF1) and YTHDC1 antibodies.
Source data are available for this figure.

regulates PRMT5 to ensure proper splicing of genes essential for cancer cell proliferation (Koh et al, 2015). Given that the ST-MYCL–EP400 complex selectively activates a subset of MYC target genes (Cheng et al, 2017), we examined whether it regulates PRMT5 expression. We also investigated whether PRMT5 inhibition affects its own protein levels. However, PRMT5 protein levels remained unchanged after *EP400* knockdown or PRMT5 inhibition,

suggesting that PRMT5 functions upstream of the Tip60-EP400 complex and is not directly regulated by it (Fig 5B).

Interestingly, increasing doses of PRMT5 inhibitor correlated with an enhanced γ-H2AX signal, indicative of DNA damage, whereas *EP400* knockdown reduced this effect (Fig 5B). This observation supports the model in which PRMT5 influences the DNA damage response, potentially in a Tip60-EP400 complex–dependent manner.

This aligns with the established role of Tip60, which is recruited to double-strand breaks to acetylate ATM at K3016, thereby activating its kinase function and initiating downstream phosphorylation cascades, including γ-H2AX formation (Sun et al, 2007).

### m6A-dependent splicing mechanism

Given the well-established role of m6A modification in splicing regulation, such as the m6A reader YTHDC1 recruiting SRSF proteins to m6A-enriched sites near splice junctions to promote exon inclusion (Xiao et al, 2016), we investigated whether PRMT5 influences the interaction between SRSF1 and YTHDC1. Using an MKL-1 stable cell line with Dox-inducible HA-tagged SRSF1 expression, we immunoprecipitated YTHDC1 and assessed its binding to SRSF1. We observed a dose-dependent reduction of HA-SRSF1 coimmunoprecipitated with YTHDC1 upon treatment with increasing concentrations of JNJ-64619178 (Fig 5C). These results indicate that the YTHDC1-SRSF1 interaction is PRMT5-dependent and potentially relies on SDMA modification of SRSF1. A parallel requirement was observed in AML, where loss of PRMT5-mediated SDMA on SRSF1 disrupted cassette-exon inclusion genome-wide (Radzisheuskaya et al, 2019).

This proposes a model in which m6A modifications near weak splice sites recruit YTHDC1 and SRSF1 to promote exon inclusion and intron removal, ensuring precise splicing fidelity. Mechanistically, PRMT5 may regulate this process by methylating SRSF1, facilitating its recruitment to regions near weak splice sites to maintain proper splicing of full-length Tip60, the catalytic subunit of the ST-MYCL-EP400 complex, thereby sustaining MCPyV ST-driven transcriptional activity.

### PRMT5 inhibition does not alter expression or splicing of MCPyV T antigen transcripts

Given the observed down-regulation of E2F target and G2/M checkpoint genes known to be activated by the Merkel cell polyomavirus (MCPyV) T antigen oncogene, we investigated whether PRMT5 inhibition affected the expression or alternative splicing of MCPyV T antigen transcripts. Analysis of both small and large T antigen transcripts revealed no significant differences in expression levels or splicing patterns between PRMT5-inhibited and control samples (adjusted $P <$ 0.05; Table S9). These results suggest that the effects of PRMT5 inhibition on cell cycle–related pathways are unlikely to be mediated by changes in MCPyV T antigen transcript regulation.

## Discussion

How PRMT5 deficiency selectively affects a subset of splice sites while sparing others remains poorly understood. Braun et al provide evidence that in both human glioblastoma cells and murine neuronal progenitor cells, PRMT5 inhibition primarily disrupts splicing at detained introns, a distinct class of retained introns, yet exerts little influence on canonical alternative splicing events, including skipping of cassette exons (Braun et al, 2017). However, this finding contrasts with our long-read sequencing results in MCC cells, where skipping of cassette exons, including KAT5 exon 5, appears significantly affected (Fig 3B). Whether this discrepancy arises from cell-type differences or from variation in sequencing approaches (e.g., short-read versus long-read) remains an open question. Braun et al (2017) also propose that PRMT5 inhibition disproportionately affects introns harboring weak 5′ splice sites, thereby promoting their retention; conversely, exon skipping and other alternative splicing events seem less reliant on PRMT5 and thus remain largely intact. However, this model does not explain why numerous detained introns and most cassette exons remain unaffected by PRMT5 deficiency, even when they possess weak 5′ splice sites. Indeed, Braun et al (2017) hypothesize that given PRMT5's role in Sm protein methylation and snRNP assembly, introns with the "weakest" recognition sequences would be the first to show deficits once PRMT5 deficiency depletes snRNA pools. Yet, the absence of a detectable reduction in snRNA levels or nuclear localization does not support that hypothesis. Braun et al (2017) further demonstrate that knocking down PRMT5 cofactors, CLNS1A or RIOK1, modulates cellular sensitivity to the PRMT5 inhibitor EPZ015666. Specifically, CLNS1A directs PRMT5 activity toward snRNP methylation and splicing, so its depletion increases sensitivity to the inhibitor. In contrast, RIOK1 directs PRMT5 activity toward ribosomal biogenesis, so its depletion confers resistance to PRMT5 inhibition. Despite the CLNS1A/RIOK1 ratio serving as a useful predictive biomarker for PRMT5 inhibitor sensitivity, it still cannot explain the selective nature of splicing deregulation upon PRMT5 deficiency. It is therefore highly likely that other PRMT5-methylated splicing regulators, which interact with exonic and intronic enhancer/silencer elements near splice sites, cooperate with weak 5′ splice sites to dictate selective splicing deregulation, a hypothesis we explore in the present study.

The extensive array of hnRNP isoforms detected in MCC cells indicates frequent alternative splicing of hnRNP transcripts themselves, reflecting a scenario where spliceosomes repeatedly select among competing splice sites rather than following a constitutive pathway. Typically, these cassette-exon decisions arise from competition between exon-enhancing SRSF proteins and exon-silencing hnRNPs, a balance precisely tuned by post-translational modifications (PTMs), such as phosphorylation and PRMT-mediated methylation. Thus, the observed hnRNP isoform diversity indirectly suggests an active, PTM-regulated interplay between these protein families. We therefore propose that under PRMT5 deficiency, reduced SDMA levels weaken specific SRSF–hnRNP interactions, disrupting this balance and contributing to the selective splicing defects observed in our study.

This study identifies PRMT5 as a key regulator of alternative splicing in MCC. PRMT5 maintains full-length Tip60, thereby sustaining the ST-MYCL-EP400 complex that drives tumorigenesis (Fig 6A). PRMT5 inhibition induces exon skipping in KAT5 and alters both the composition and the histone acetyltransferase activity of the Tip60-EP400 complex (Table 1; Fig S1G and H). Consistent with its role in preventing exon skipping in MYC-driven cancers, PRMT5 likely promotes SRSF1 recruitment to Tip60 pre-mRNA to support splicing of full-length, catalytically active Tip60. This process depends on m6A-modified regions near splice sites, where PRMT5-modified

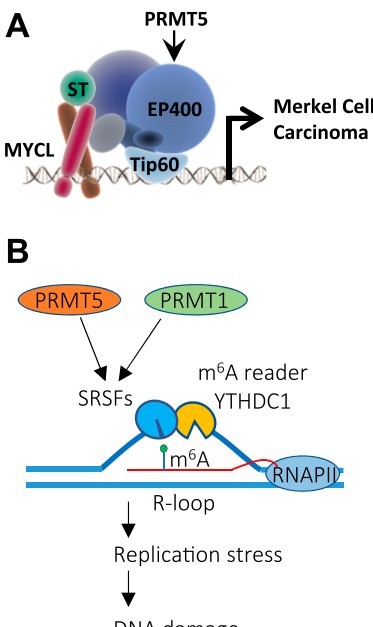

**Figure 6. Model of PRMT5 function in Merkel cell carcinoma and in limiting DNA damage.**
**(A)** PRMT5 promotes exon inclusion to maintain full-length Tip60, sustaining activity of the ST-MYCL-EP400 complex, a major driver of Merkel cell carcinoma tumorigenesis. **(B)** PRMT5 methylates SRSF proteins, enhancing their recruitment to m6A-marked RNA near splice sites that can coincide with R-loops. This PRMT5-SRSF axis ensures proper splicing and promotes R-loop resolution, thereby reducing replication stress and limiting DNA damage.

SRSF1 interacts with YTHDC1, an m6A reader involved in RNA splicing regulation. PRMT5 inhibition disrupts this interaction, leading to exon skipping, altered Tip60 activity, and downstream effects on genome stability. R-loops are three-stranded nucleic acid structures formed when RNA hybridizes to its complementary DNA strand, and often overlap with m6A-modified sites (Yang et al, 2019; Abakir et al, 2020). Although R-loops play regulatory roles in transcription, their excessive accumulation can lead to replication stress and DNA damage. Splicing deregulation, particularly intron retention, increases R-loop formation by generating unspliced RNA that remains hybridized to DNA, preventing proper resolution. Thus, PRMT5-modified SRSF proteins, by facilitating correct splicing at splice sites near m6A-enriched regions, can potentially prevent unscheduled R-loop formation and suppress the activation of DNA damage pathways (Fig 6B). To rigorously test this model, future studies should generate an unmethylatable SRSF1 mutant by substituting key SDMA-acceptor arginines with lysines. Introducing this mutant into PRMT5-inhibited or PRMT5-deficient MCC cells would clarify whether the loss of PRMT5-dependent methylation disrupts the binding of SRSF1 to the m6A reader YTHDC1 and consequently fails to restore *KAT5* exon 5 inclusion or other PRMT5-sensitive splicing events. Demonstrating that WT SRSF1, but not the unmethylatable variant, rescues YTHDC1 interaction and corrects splicing would provide direct causal evidence linking PRMT5-mediated methylation to m6A-dependent splice-site recognition.

Building on the pathway enrichment analysis (Fig 2D), we next consider how p53 activation and MYC signaling might intersect with

PRMT5 dependency in MCC. *TP53* is the most frequently mutated tumor suppressor in human cancer. Several tumor viruses, including polyomaviruses, papillomaviruses, and adenoviruses, express viral tumor antigens that bind to and inactivate p53. However, MCPyV isolated from tumor specimens always carries a truncated large T antigen because of integration into the host genome. Consequently, it does not bind p53, suggesting alternative mechanisms of WT p53 suppression in virus-positive MCC (Cheng et al, 2013). Our data suggest that PRMT5 contributes to p53 suppression. PRMT5 has also been shown to function as a MYC corepressor, repressing MYC targets such as p21, a key regulator of cell cycle progression. The identification of MYC targets as the most up-regulated pathway in our analysis suggests functional differences between *MYC* and its paralog *MYCL*. We previously demonstrated that *MYCL*, but not *MYC* or *MYCN*, is expressed in virus-positive MCC cell lines (Cheng et al, 2017). Our data strongly suggest that reactivating MYC targets suppressed by PRMT5 may be a potential strategy to induce MCC cell death. Therefore, it would be important to investigate in future studies whether inhibiting p53 activation can desensitize MCC cells to PRMT5 deficiency or whether overexpressing *MYC* is synergistically lethal with PRMT5 inhibition.

Although the precise cell of origin for MCC remains uncertain, our findings suggest that MCPyV may preferentially infect cells with inherently high PRMT5 levels to drive oncogenic transformation. PRMT5 ensures proper splicing of full-length, catalytically active Tip60 (Tip60α), a critical driver of MCC tumorigenesis and tumor maintenance. Beyond its role in splicing, PRMT5's association with maintaining stem-like properties in cancer cells (Braun et al, 2017) reinforces the concept that elevated PRMT5 activity creates a favorable cellular environment for MCPyV-driven oncogenesis. Consequently, PRMT5 represents a compelling therapeutic target in MCC.

Our data reveal a critical link between PRMT5 activity and the DNA damage response (DDR). PRMT5 inhibition leads to γ-H2AX accumulation, suggesting that exon skipping in DDR pathway genes disrupts their proper splicing regulation, impairing the cellular response to DNA damage. In particular, Tip60, a key factor in chromatin remodeling and DNA repair, is significantly impacted by PRMT5 inhibition. Consistent with this, *EP400* knockdown partially rescues γ-H2AX accumulation, demonstrating that PRMT5 influences DDR in an EP400-dependent manner. Given its dual role in oncogenic transcription and DDR regulation, targeting PRMT5 in MCC may simultaneously inhibit tumor growth and enhance sensitivity to DDR-targeting agents. Thus, combining PRMT5 inhibitors with DDR-targeting drugs could significantly improve therapeutic efficacy. Furthermore, our data indicate that PRMT5 regulates antigen presentation pathways (Fig 3C), suggesting potential immunotherapeutic benefits. Further studies should explore whether PRMT5-targeting strategies could enhance immune checkpoint blockade therapy by restoring antigen presentation in MCC tumors.

Given PRMT5's central role in splicing regulation, further studies are needed to identify additional PRMT5-methylated splicing factors. Defining the full spectrum of PRMT5 substrates through SDMA immunoprecipitation followed by large-scale mass spectrometry, along with cross-linking immunoprecipitation followed by high-throughput sequencing (CLIP-seq) analysis of potential RBPs, would help expand

**Table 2.  Antibody details for immunoblot analysis.**

| Antibody | Host species | Dilution | Vendor | Catalog number |
|---|---|---|---|---|
| EP400 | Rabbit | 1:5,000 | Bethyl Laboratories | A300-541A |
| PRMT5 | Rabbit | 1:5,000 | Cell Signaling Technology | 79998 |
| HA | Mouse | 1:5,000 | Cell Signaling Technology | 2367 |
| SDMA | Rabbit | 1:5,000 | Cell Signaling Technology | 13222 |
| ADMA | Rabbit | 1:5,000 | Cell Signaling Technology | 13522 |
| MMA | Rabbit | 1:5,000 | Cell Signaling Technology | 8015 |
| GAPDH | Rabbit | 1:5,000 | Cell Signaling Technology | 5174 |
| γH2Ax | Rabbit | 1:5,000 | Cell Signaling Technology | 2577 |
| YTHDC1 | Rabbit | 1:5,000 | Cell Signaling Technology | 77422 |
| H4K5ac | Rabbit | 1:5,000 | Cell Signaling Technology | 8647 |
| H4K12ac | Rabbit | 1:5,000 | Cell Signaling Technology | 13944 |
| H4K16ac | Rabbit | 1:5,000 | Cell Signaling Technology | 13534 |
| H2A.Zac | Rabbit | 1:5,000 | Cell Signaling Technology | 75336 |
| H3 | Rabbit | 1:5,000 | Cell Signaling Technology | 4620 |
| Anti-rabbit IgG, HRP-conjugated | Goat | 1:5,000 | Cell Signaling Technology | 7074 |
| Anti-mouse IgG, HRP-conjugated | Horse | 1:5,000 | Cell Signaling Technology | 7076 |
| Tip60 (C-7) | Mouse | 1:5,000 | Santa Cruz Biotechnology | sc-166323 |
| MAX (C-17) | Rabbit | 1:5,000 | Santa Cruz Biotechnology | sc-197 |
| Vinculin | Mouse | 1:30000 | MilliporeSigma | V9131 |
| ASYM24 | Rabbit | 1:5,000 | MilliporeSigma | 07-414 |
| ASYM25 | Rabbit | 1:5,000 | MilliporeSigma | 09-814 |

List of antibodies used in immunoblot experiments, with host species, working dilution, vendor, and catalog information.

the catalog of PRMT5-methylated RBPs and validate their binding motifs near splice sites. This may also uncover additional RNA motifs or modifications beyond m6A that contribute to PRMT5-driven splicing regulation. Furthermore, YTHDC1 immunoprecipitation followed by large-scale mass spectrometry could identify additional PRMT5-methylated RBPs beyond SRSF1, further elucidating their roles in PRMT5-mediated splicing regulation.

# Materials and Methods

### Immunoblot analysis

Cells were lysed in RIPA buffer supplemented with protease and phosphatase inhibitors. Protein concentration was determined by the Bradford assay. Lysates were mixed with 2x Laemmli sample buffer and heated at 95°C for 5 min, and 30–60 µg of whole-cell lysates per lane was resolved on 4–20% Criterion gradient SDS–PAGE gels (Bio-Rad). Proteins were then transferred to PVDF membranes using CAPS buffer (10 mM CAPS (3-[cyclohexylamino]-1-propanesulfonic acid), pH 10.5, with 15% methanol) and blocked in 1x KPL blocking solution (SeraCare). Membranes were incubated overnight at 4°C with primary antibodies diluted in 0.05x KPL solution, washed in 1x TBST solution (20 mM Tris, 150 mM NaCl, 0.1% Tween-20, pH 7.4) five times for 5 min each, incubated with HRP-conjugated secondary antibodies in 0.05x

KPL solution for 1 h at RT, and washed again in 1x TBST five times for 5 min each. Signals were developed with Clarity MAX ECL substrates (Bio-Rad) and imaged on Bio-RadV3 ChemiDoc Imager. Band intensities were quantified in ImageJ; for whole-cell lysates, targets were normalized to GAPDH or vinculin, and for histone blots to histone H3. Uncropped blots and replicate images are provided in Supplemental Data 1 file. Antibody details are listed in Table 2.

### Immunoprecipitation (IP)

Cells were lysed in EBC lysis buffer (50 mM Tris–HCl, pH 8; 150 mM NaCl; 0.5% NP-40; 0.5 mM EDTA, pH 8; 1:10,000 $\beta$-mercaptoethanol) supplemented with protease and phosphatase inhibitors. After protein quantification using the Bradford assay, 1 mg of cell lysate was incubated with 1 µg of primary antibody and 15 µl of magnetic Protein G Dynabeads (Invitrogen) at 4°C overnight with head-to-head rotation. Beads were washed with high-salt EBC buffer (50 mM Tris–HCl, pH 7.4; 300 mM NaCl; 0.5% NP-40; 0.5 mM EDTA, pH 8) and subsequently processed for immunoblot analysis.

### CellTiter-Glo viability assays

MKL-1 cells were treated with AccuMax at RT for 5 min to dissociate cell clumps and counted using a hemocytometer. A total of

3,000 cells per well were plated in 96-well plates and treated with the respective inhibitors. Cell viability was assessed using the CellTiter-Glo assay (Promega), which lyses cells to measure ATP levels, reflecting metabolically active and viable cells. Luminescence was detected using a GloMax luminometer plate reader (Promega).

### Calcein AM + EthD-1 live/dead cell assays

MKL-1 cells, pretreated with either DMSO or inhibitors, were dissociated with AccuMax to reduce clumping and then diluted to $1 \times 10^6$ cells/ml in a 500 $\mu$l volume. Calcein AM was added to a final concentration of 0.01 $\mu$M and incubated for 15 min, followed by ethidium homodimer-1 (EthD-1) at a final concentration of 100 ng/ml for another 15 min. A 10 $\mu$l aliquot of the stained cells was then placed into each well of a Cellvis 24-well glass-bottom black plate and covered with a round, 15-mm microscope coverslip. Fluorescence images were acquired using Invitrogen EVOS Imaging System.

### Tissue culture

HEK293T cells were obtained from the ATCC. The MCC line MKL-1 was obtained from Sigma-Aldrich. Two MKL-1 derivatives were generated for this study: a doxycycline-inducible EP400 shRNA line and a doxycycline-inducible HA-tagged SRSF1 line used for YTHDC1 coimmunoprecipitation. Cloning details are provided in **Plasmid Constructs**, and lentiviral production in HEK293T cells and generation of the stable MKL-1 derivatives are described in **Lentiviral Transduction and Stable Cell Line Generation**. Cells were maintained in RPMI 1640 supplemented with 10% fetal bovine serum, 1% GlutaMAX, and 1% penicillin and streptomycin. Cells were passaged at 0.5 million cells per milliliter. HEK293T cells were passaged every 3–4 d. MKL-1 suspension cultures were passaged once per week; cultures received one feed between passages by adding fresh medium to the flask. Early passage cells were expanded to create master and working banks in liquid nitrogen. For experiments, vials were thawed from the working bank. HEK293T and MKL-1 cultures were used within a maximum of 25 passages after thaw. *Mycoplasma* testing by PCR was performed routinely and was negative.

### Plasmid constructs

The EP400 shRNA sequence (GCAAAGACATCCACATATACA ctcgag TGTA-TATGTG GATGTCTTTG C ttttt) was cloned into the Tet-pLKO-puro vector (plasmid #21915; Addgene) between AgeI and EcoRI restriction sites. The SRSF1 isoform 1 ORF was obtained from the Harvard University Plasmid Information Database (PlasmID), PCR-amplified, and fused with a 3×HA tag via PCR. The resulting construct was cloned into the Dox-inducible lentiviral vector pLIX_402-puro (plasmid #41394; Addgene) using Gateway cloning.

### Lentiviral transduction and stable cell line generation

HEK293T cells were used as packaging cells for lentivirus production. For each transfection, HEK293T cells were plated in a 60-mm dish at ~70% confluency. Then, 7 $\mu$g of lentiviral expression vector, 5 $\mu$g of packaging plasmid (plasmid #12260, psPAX2; Addgene), and 2 $\mu$g of envelope plasmid (#12259, pMD2.G [VSV-G]; Addgene) were mixed with 42 $\mu$l of poly-ethylenimine (PEI, Polysciences) and cotransfected into the cells. Two days post-transfection, the HEK293T cell supernatant was filtered (0.45 $\mu$m) and supplemented with 2 $\mu$g/ml polybrene before transducing recipient cells. Spinoculation was performed at 931$g$ for 2 h at RT to enhance transduction efficiency. After centrifugation, fresh media were added to reduce polybrene concentration to 1 $\mu$g/ml, followed by a medium change the next day to remove residual polybrene. To generate stable cell lines, cells were selected with 1–2 $\mu$g/ml puromycin for 5 d.

### Preparation of Illumina RNA-sequencing libraries and sequencing

Total RNA was extracted from biological samples using RNeasy Plus Mini Kit (QIAGEN). RNA quality and quantity were assessed using a NanoDrop spectrophotometer, ensuring OD260/280 ≥ 2.0 and OD260/230 ≥ 2.0. Samples were submitted to Novogene for Illumina RNA sequencing. Briefly, strand-specific libraries were prepared using the NEBNext Ultra II Directional RNA Library Prep Kit, following the manufacturer's instructions. ≥400 ng of total RNA (measured by Qubit) was used. mRNA was purified from total RNA using poly-T oligo-attached magnetic beads. After fragmentation, first-strand cDNA was synthesized using random hexamer primers, followed by second-strand cDNA synthesis with dUTP instead of dTTP to maintain strand specificity. The strand-specific libraries then underwent end repair, A-tailing, adapter ligation, size selection, amplification, and purification. Size selection was performed using AMPure XP beads to isolate fragments of ~300 bp. Library quality was assessed using Qubit and real-time PCR for quantification and Bioanalyzer for size distribution analysis. Equimolar amounts of each quantified library were pooled for sequencing on the Illumina NovaSeq X Plus Series platform, generating paired-end 150-bp reads with 18 GB of raw data per sample (~60 million reads).

### RNA-seq differential expression analysis

We sequenced eight samples (four controls treated with vehicle DMSO, and four treated with the PRMT5 inhibitor JNJ-64619178) using both short-read sequencing (generated with Illumina technology) and full-length isoform sequencing (generated with PacBio Iso-Seq technology). Short-read sequencing data were primarily used to analyze DEGs, whereas full-length isoform data were used to analyze isoform expression, alternative splicing, and novel isoforms. The total number of reads generated for each platform and data availability accessions are summarized in Table S1.

Raw sequencing reads were examined for quality using FASTQC (Andrews, 2010). Adapters and low-quality bases were trimmed using FastP (Chen et al, 2018). Reads with a quality score below 20 and reads shorter than 30 bp after trimming were discarded. Alignment of the reads against the human reference genome (Human hg38 Gencode v39) cDNA was completed using salmon v1.10.3 with the options "--validateMappings," "-l A," and the

"--gcBias" flag (Patro et al, 2017). Salmon provides estimated counts per transcript and effective transcript length. Transcript abundance was then aggregated to the gene level based on the Gencode annotations using tximport (Soneson et al, 2015), and R v4.4. DESeq2 v1.46.0 was used for differential expression analysis with the default model, which internally corrects for library size (Love et al, 2014). To reduce the influence of low-abundance genes and improve the reliability of differential expression results, we filtered out genes with fewer than 10 counts in fewer than four samples, retaining only genes that met this threshold across at least three samples. To assess the significance of differential expression between the two conditions, we performed the Wald test, which tests for differences in log fold changes. We applied log fold change (LFC) shrinkage using the apeglm to reduce bias in effect size estimates and aid in visualization and interpretation (Zhu et al, 2019). The results were adjusted for multiple testing using a significance threshold ($\alpha$) of 0.05.

Illumina RNA sequencing was also analyzed using the rMATS-4.3.0 tool. rMATS classifies different splicing events and performs differential analysis of alternative splicing across samples (Shen et al, 2014). rMATS identifies the following alternative splicing (AS) events: Skipped Exon (SE), Mutually Exclusive Exon (MXE), Alternative 5′ Splice Site (A5SS), Alternative 3′ Splice Site (A3SS), and Retained Intron (RI). Each alternative splicing event generates two isoforms. To detect differentially expressed isoforms, expression levels are calculated by normalizing the effective transcript length and computing the ratio of exon inclusion to total isoform expression. Significant differential alternative splicing events are identified using FDR < 0.05. To visualize alternative splicing events, isoforms are aligned using the GFF annotation provided as input. Junction reads are shown as arcs connecting spliced exons, where arc thickness represents the number of junction reads. The vertical coordinate represents a modified RPKM value. rMATS output files were uploaded to the rMAPS2 website to map binding motifs (Hwang et al, 2020).

### GSEA

To identify enriched biological pathways and gene sets, we performed GSEA (Subramanian et al, 2005) using the DESeq2 results from the Illumina gene-level differential expression analysis. We used a ranked list of genes based on the Wald statistics, which is the ratio of the $\log_2$ fold change (logFC) and standard error. For GSEA, we use the hallmark gene sets, "h.all.v2024.1.Hs.symbols.gmt," from the MSigDB database, which were analyzed with 1,000 permutations (Liberzon et al, 2011). Pathways were assessed for enrichment in either up-regulated or down-regulated conditions, with significance determined using FDR < 0.25 and nominal $P$-value thresholds of < 0.01 and < 0.05.

A hypothesis-driven GSEA was also performed using a curated set of pathways and genes obtained from Lomics.ai (Wong et al, 2024 Preprint). The analysis was designed to investigate the mechanism by which PRMT5 influences gene regulation and splicing. The input query was "how does Protein Arginine Methyltransferase 5 (PRMT5) influence gene expression and RNA splicing through histone and non-histone methylation?" Pathways

and gene sets were ranked using a large language model (Meta Llama 3-70B-Instruct) with a random seed for unbiased selection. A set of 40 pathways, each with 40 genes, were obtained from Lomics.ai. GSEA was performed on the curated gene sets using the ranked list described above. The AI-driven tool helps to enable more targeted insights, reducing the FDR by focusing on context-specific biological mechanisms.

### Preparation of PacBio full-length isoform sequencing libraries and sequencing

RNA samples were extracted using the QIAGEN RNeasy plus mini kit (QIAGEN), and total RNA content was measured using a Qubit fluorometer with the RNA HS assay. Samples were run on Agilent's TapeStation 4200 to check that their RIN values were ≥7.0. The recommended input for the Kinnex protocol is at least 300 ng per library; each sample was diluted 1:5 before library prep because of high concentration of RNA in each sample. All RNA samples were synthesized into cDNA, amplified, and barcoded using the PacBio ISO-EXPRESS 2.0 kit. The DNA concentration was measured, using Qubit 1x HS dsDNA, and samples were normalized into a single pool in the Kinnex PCR. There are eight different primers for the Kinnex PCR step that label each end of the fragment to facilitate ligation and formation of orientation-specific sites. The sample pool was divided into eight PCRs, and then, all PCR products were repooled. The pool of Kinnex PCR fragments were ligated at the segment sites, using the Kinnex-specific ligase, to form a linear Kinnex cDNA product. The final pool, before SMRTlink Sample setup, was nuclease-treated and cleaned, and a 1:5 dilution of the pool was used for a final QC check. The final pool's concentration was measured using a Qubit fluorometer with the 1x HS dsDNA assay, and fragment sizes were assessed by gel-electrophoresis using Agilent's Femtopulse. The concentration and fragment sizes were used to curate the SMRTlink Sample Setup protocol that calculates the sequencing polymerase required for the final loading pool. The final loading pool used the Sequel II Binding Kit 3.2's polymerase and was sequenced on a single SMRT cell. Sequencing was completed on a PacBio Sequel IIe system with an 8 M SMRT cell. HiFi reads were generated with the PacBio SMRTlink v13.1 application using the circular consensus sequencing (CCS) algorithm.

### Isoform discovery and classification

Kinnex PacBio HiFi reads were processed to obtain high-quality isoforms and classify them into structural categories. The reads were segmented using the PacBio SKERA "split" utility, targeting the standard Kinnex adapters. "pbmerge" was then used to combine the two PacBio runs into a single BAM file. The combined segmented reads were demultiplexed into samples using Lima with the options "--ignore-biosamples --isoseq --peek-guess --overwrite-biosample-names." Lima identifies and removes barcoded Iso-Seq primer sequences and writes a BAM file per sample. After demultiplexing, Iso-Seq v4.2.0 was used for isoform discovery and characterization. Iso-Seq "refine" with default settings was used to trim poly(A) tails and remove concatemers. De novo isoform-level clustering was completed using Iso-Seq "cluster2" with default settings. The clustered reads were then

mapped to the human reference genome (Human hg38 Gencode v39) using PacBio's implementations of the minimap2 aligner with the "ISOSEQ" preset for alignment and the option to sort output (Li, 2016). Unique isoforms were identified using Iso-Seq "collapse" with the option "--do-not-collapse-extra-5exons", which generated a collapsed GFF file of isoforms.

Isoform classification and filtering were performed using PacBio Pigeon application. Reference files for the human genome (Human hg38 Gencode v39) were obtained from PacBio's publicly available FTP site (https://downloads.pacbcloud.com/public/dataset/MAS-Seq/REF-pigeon_ref_sets/Human_hg38_Gencode_v39/); this includes a BED formatted file of CAGE (Cap Analysis of Gene Expression) peaks with information on transcription start sites (TSS) across the genome, Intropolis data with information on splice junction sites, and a list of polyA motifs. The GFF file from Iso-Seq "collapse2" was sorted by position with Pigeon "sort" and classified into structural categories using Pigeon "classify" with default settings. Pigeon uses the SQANTI isoform classification system (Pardo-Palacios et al, 2024) to categorize transcript models based on their splice junction concordance with known reference annotations. Briefly, "Full Splice Match" (FSM) isoforms exactly match a known transcript's splice junctions; "Incomplete Splice Match" (ISM) isoforms share internal splice sites with known transcripts but lack complete exon coverage. Isoforms classified as "Novel in Catalog" (NIC) use only annotated splice sites but in new combinations, whereas "Novel Not in Catalog" (NNC) incorporates at least one previously unannotated donor or acceptor site. Additional categories include "Antisense" (overlapping an annotated gene on the opposite strand), "Genic Intron" (contained entirely within a known intron), "Genic Genomic" (overlapping both intronic and exonic regions), and "Intergenic" (located outside annotated gene boundaries). Isoforms with artifacts, such as mono-exonic isoforms, those with evidence of reverse transcriptase template switching, intrapriming, or isoforms with low coverage at non-canonical splice sites, were filtered using Pigeon "filter" with default settings. The Pigeon "report" tool was used to generate read subsampling reports at 5,000-read intervals.

A total of 47,936,880 PacBio HiFi reads with >99% accuracy were generated. After quality filtering and preprocessing, 46,477,422 reads passed quality control thresholds and refinement (mean per sample: 5,819,382; SD: 613,324). These were clustered into 43,897,431 full-length reads, ultimately collapsing into 875,968 unique isoforms. After structural classification with SQANTI (Pardo-Palacios et al, 2024), 374,822 isoforms (42.79% of all detected isoforms) were removed as likely artifacts. Within this filtered set, intrapriming accounted for 32.55%, reverse transcriptase switching for 2.18%, and low coverage or non-canonical features for 65.27%. A total of 501,147 isoforms remained after filtering, 25.66% of which are classified as a full-splice match (FSM) and 68.82% are classified as novel isoforms. Novel isoforms are further divided into incomplete splice match (ISM; 32.61%), novel in catalog (NIC; 18.89%), novel not in catalog (NNC; 17.32%), antisense (0.10%), genic genomic (0.02%), intergenic (0.05%), and "other" (0.07%). These isoforms were subsequently assigned to 79,602 genes (20,457 known and 59,145 novel) and 428,260 unique transcripts. Gene-level saturation curves (Fig

S4) were generated to ensure that the sequencing depth was adequate for the analysis. Table S5 provides a breakdown of the isoform structural categories and associated gene names. The observed proportion of novel isoforms (68.82%) is consistent with previous Iso-Seq studies, which have reported similarly high levels of novel transcript discovery, ranging from 77% (Glinos et al, 2022) to 82.9% (Wang et al, 2024 *Preprint*) in human samples.

### Differential expression analysis of full-length isoform data

Differential gene and isoform analysis was carried out using DESeq2, using the read counts produced from Iso-Seq. For comparison with the Illumina data, we followed the same analysis outlined above with a few adjustments; read counts were first aggregated at the gene level, and lowly abundant genes were filtered out if they had fewer than five counts in fewer than three samples. Thresholds for count filtering were slightly adjusted for full-length isoform data to account for its typically lower depth of coverage compared with the short-read dataset. To assess the correlation between gene expression measurements obtained from PacBio and Illumina sequencing, we computed gene-wise average expression values for each platform. For full-length isoform data, counts per million (CPM) were calculated and log-transformed, whereas for short-read data, variance-stabilized transformation (VST) was applied. A Pearson correlation analysis was performed to quantify the relationship between the two datasets. We visualized the correlation using a scatter plot, where each point represented a gene, and a linear regression line was fitted to illustrate the overall trend. To summarize gene-level expression changes while accounting for isoform variability, we calculated a weighted mean LFC for each gene, where individual isoform LFCs were weighted by their baseMean (average normalized expression). Genes were classified as up-regulated, down-regulated, or nonsignificant based on a threshold-adjusted false discovery rate (FDR), *P*-value (*P*adj), and LFC. Isoform counts per gene were also determined to assess gene expression variability.

### Comparison of isoform structural categories across DEGs

To compare the distribution of up-regulated and down-regulated genes across different isoform structural categories in the dataset, a chi-squared test for independence was performed. For each category, the observed counts of up-regulated and down-regulated genes were compared with the expected counts, which were based on the overall distribution of up-regulated and down-regulated genes across all categories. The chi-squared test was applied using the chi2_contingency function from the Python3 scipy.stats package. We then used gProfiler to investigate pathways and gene sets that are significantly enriched across structural categories, focusing on intron retention and exon skipping events (Kolberg et al, 2023). The Benjamini–Hochberg method was used for FDR calculation, a minimum and maximum functional category size was set to 3 and 500 respectively,

and the databases were limited to include "Gene Ontology Biological Process," "Reactome," and "KEGG." An enrichment map visualization showing functionally related gene sets was generated using Cytoscape v3.10.3 (Shannon et al, 2003) with the EnrichmentMap plugin (Merico et al, 2010). Enriched pathways with FDR < 0.05 were included in the network, with edges representing gene set similarity based on shared genes. Clusters of related pathways were identified and annotated using the AutoAnnotate plugin (Kucera et al, 2016).

### Statistics

Routine statistical tests and custom scripts were completed in R v2024.09.1+394 and Python v3.6.9. *$P < 0.05$, **$P < 0.01$, ***$P < 0.001$, ****$P < 0.0001$. Specific statistical tests are outlined in the figure or table legends.

## Data Availability

The datasets and computer code produced in this study are available in the following databases:

- Illumina RNA-seq short-read data: Gene Expression Omnibus GSE290105
- PacBio Iso-Seq full-length isoform data: Gene Expression Omnibus GSE290569
- Computer scripts and code: GitHub (https://github.com/Joseph7e/PRMT5_MCC_alt-splicing)

## Supplementary Information

## Acknowledgements

This publication was supported by the NH-INBRE program and the Center for Integrated Biomedical and Bioengineering Research (CIBBR) through grants from the National Institute of General Medical Sciences of the National Institutes of Health under Award Numbers P20GM103506 and P20GM113131, respectively.

### Author Contributions

JL Sevigny: conceptualization, data curation, software, formal analysis, visualization, methodology, and writing—original draft, review, and editing.
EV Proctor: data curation, validation, investigation, and methodology.
H Adams: validation, investigation, and methodology.
ER MacDonald: data curation and methodology.
WK Thomas: conceptualization, resources, data curation, software, supervision, and methodology.
J Cheng: conceptualization, resources, data curation, software, formal analysis, supervision, funding acquisition, validation, investigation, visualization, methodology, project administration, and writing—original draft, review, and editing.

### Conflict of Interest Statement

The authors declare that they have no conflict of interest.

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
