## [Reviewer comments · Life Science Alliance]

Life Science Alliance

Protein Arginine Methyltransferase 5 Sustains Tip60-EP400 Complex via SRSF1 in Merkel Cell Carcinoma

Joseph Sevigny, Evelyn Proctor, Heather Adams, Erin MacDonald, W Thomas, and Jingwei Cheng
DOI: <https://doi.org/10.26508/lsa.202503316>

Corresponding author(s): Jingwei Cheng, University of New Hampshire

Review Timeline:

Submission Date:	2025-03-19
Editorial Decision:	2025-05-02
Revision Received:	2025-07-02
Editorial Decision:	2025-07-31
Revision Received:	2025-08-06
Editorial Decision:	2025-08-08
Revision Received:	2025-08-10
Accepted:	2025-08-12

Scientific Editor: Sarita Hebbar

Transaction Report:

May 2, 2025

Re: Life Science Alliance manuscript #LSA-2025-03316-T

Dr. Jingwei Cheng
University of New Hampshire
Molecular, Cellular and Biomedical Sciences
46 College Road
Rudman 206
Durham, NH 03824

Dear Dr. Cheng,

Thank you for submitting your manuscript entitled "PRMT5-Dependent SRSF1 Modification Promotes Tip60-EP400 Complex Activity via Exon Inclusion" to Life Science Alliance. The manuscript was assessed by two expert reviewers, whose comments are appended to this letter.

Overall the reviewers find this work of significance. Having said that, we agree with the reviewers that the manuscript needs to be revised. The revised manuscript must include data on the reproducibility of the rescue of splicing defects by low-dose SRSF1 over expression (as suggested by Reviewer 1). Additionally, the results must be further elaborated upon, with changes to the text especially when referring to TipE60 isoforms described in this work (Reviewer 1, point 12) and in the context of published literature (Reviewer 2, second para, point 3).

We invite you to submit a revised manuscript addressing the Reviewer comments. When submitting the revision, please include a letter addressing all the reviewers' comments point by point. While a rebuttal must respond to all points in some form, additional data to resolve these points is not required. The typical timeframe for revisions is three months. Please note that papers are generally considered through only one revision cycle, so strong support from the referees on the revised version is needed for acceptance

You will be guided to complete the submission of your revised manuscript and to fill in all necessary information. Please get in touch in case you do not know or remember your login name. While you are revising your manuscript, please also attend to the below editorial points to help expedite the publication of your manuscript. Please direct any editorial questions to the journal office.

We hope that the comments below will prove constructive as your work progresses. Thank you for this interesting contribution to Life Science Alliance. We are looking forward to receiving your revised manuscript.

Sincerely,

Sarita Hebbar, PhD
Scientific Editor
Life Science Alliance
<http://www.lsjournal.org>

B. MANUSCRIPT ORGANIZATION AND FORMATTING:

Reviewer #1 (Comments to the Authors (Required)):

The authors demonstrate that the MCC cell line MKL-1 is sensitive to PRMT5 inhibition, leading to reduced expression of the full-length Tip60 isoform. To explore the impact of PRMT5 inhibition on splicing, they conducted both short- and long-read sequencing, comparing untreated cells with those treated with the PRMT5 inhibitor JNJ-64619178. Their analysis revealed changes in alternative splicing, particularly in intron retention and exon skipping. The authors propose a connection between Tip60 splicing changes, the function of SRSF1 in Tip60 splicing, and the potential link of SRSF1 function to m6A RNA modification.

While the findings reinforce/confirm previously established mechanisms, they offer limited novelty. Differential gene expression and splicing analysis, the impact on Tip60 splicing and cell proliferation, the dynamics of SDMA response to inhibition, and the types of alternative splicing events observed have all been reported in multiple prior studies (Braun et al., Radziskeuskaya et al., Bezzi et al., Hamard et al., Clarke et al., and others). While reproducibility in a distinct cellular context is valuable, the experimental outcomes largely confirm existing literature rather than presenting new insights. Furthermore, the established role of PRMT5 in regulating SRSF1 methylation and its impact on splicing has been documented since 2019 (<https://www.nature.com/articles/s41594-019-0313-z>), yet this study is not cited or mentioned in the manuscript. The only novel aspect of the study appears in Figures 5A and 5C but needs to be developed considerably to establish its reproducibility and relevance. The partial rescue of Tip60 splicing defects by low-dose SRSF1 overexpression may provide a mechanistic link supporting the proposed model, but this experiment needs to be reproduced and assessed statistically. The potential involvement of m6A methylation and YTHDC1 is intriguing but remains unexplored by the authors. Further mechanistic studies, such as assessing whether an unmethylatable SRSF1 mutant can rescue splicing defects or bind to YTHDC1, would enhance the study's contribution beyond existing knowledge.

Additional Comments:

1. The manuscript states that 68.82% of identified isoforms were classified as novel. Is this proportion typical for long-read RNA sequencing studies?
2. Do the genes exhibiting differential isoform expression overlap with those identified as differentially expressed?
3. The manuscript should clarify the classification of isoforms (e.g., novel in catalog vs. novel not in catalog). Similarly, the categories in Figure 3A require explanation.
4. In EV3, the cut-off values for differentially expressed genes (DEGs) should be included in the figure legend.
5. The following paragraph seems incomplete. It should explicitly state that these conditions were used for subsequent experiments:
"To ensure that the observed effects were specifically due to PRMT5 inhibition and not secondary to cell death, we established experimental conditions in which MKL-1 cells were treated with 12 nM JNJ-64619178 for fewer than five days. Under these conditions, any changes detected could be attributed to PRMT5 inhibition rather than cytotoxicity."
6. A figure reference is missing in this statement:
"In an MKL-1 stable cell line expressing doxycycline (Dox)-inducible EP400 shRNA, treatment with either Dox or 12 nM JNJ-64619178 resulted in reduced acetylation of histone H4 at lysine 5, 12, and 16, as well as decreased acetylation of the H2A.Z histone variant."
7. The manuscript states: "However, full-length EP400 protein levels were not significantly altered by PRMT5 inhibition in MKL-1 cells." Where is the supporting evidence for this claim?
8. The following paragraph belongs in the discussion section rather than the results:
"P53 is the most frequently mutated tumor suppressor in human cancer. Several tumor viruses, including polyomaviruses, papillomaviruses, and adenoviruses, express viral tumor antigens that bind to and inactivate p53... Therefore, it would be important to investigate in future studies whether inhibiting p53 activation can desensitize MCC cells to PRMT5 deficiency or if overexpressing MYC is synergistically lethal with PRMT5 inhibition."
9. Figure 2B presents selected gene names. How were these genes chosen for display?
10. The logic in the following statement is unclear:

"The substantial presence of large numbers of hnRNP isoforms in MCC cells suggests an important regulatory balance involving SRSF proteins and their post-translational modifications, such as phosphorylation or PRMT-mediated methylation." A large number of isoforms does not inherently imply a regulatory balance between PTMs. This reasoning should be better articulated.

11. The following statement lacks clarity:

"The mix of upregulated (red) and downregulated (blue) genes suggests that PRMT5 inhibition influences alternative splicing patterns."

How does a mixture of up- and downregulated genes imply a specific effect on alternative splicing?

12. Figure 4C highlights the skipped exon of Tip60, but this exon does not include the catalytic domain. The manuscript previously stated that loss of the full-length Tip60 isoform leads to a catalytically inactive protein. This apparent contradiction should be addressed.

13. Figure 4D's legend should provide a more detailed explanation of the graph and the meaning of different lines.

Conclusion:

This study reinforces known mechanisms of PRMT5-mediated splicing regulation but offers limited new insights. The findings in Figures 5A and 5C provide the most meaningful additions, yet their mechanistic underpinnings remain underexplored. Addressing the questions posed above would enhance the impact and originality of the work.

Reviewer #2 (Comments to the Authors (Required)):

The manuscript by Sevigny et al. demonstrated that PRMT5, which regulates splicing of Tip60/KAT5, is required for Merkel cell polyomavirus (MCPyV)-positive Merkel cell carcinoma (MCC) tumorigenicity. Dr. Cheng, a corresponding author of this manuscript, previously identified the MycL/Max-EP400-Tip60 histone acetyltransferase complex as one of the critical cellular oncogenic targets of the Merkel cell polyomavirus (MCPyV) small T oncoprotein (ST). In this study, his group found that one of the Tip60 isoforms, which appears to be catalytically inactive and results from exon 5 skipping, is increased by PRMT5 inhibitors. PRMT5 inhibition suppressed histone acetylation associated with the Tip60-HAT complex and suppressed proliferation of virus-positive MCC tumor cells. In addition, the authors identified SRSF1 as a regulator of Tip60 mRNA. Overall, the experiments are carefully performed and the results support their conclusion.

However, the manuscript can be improved if the authors address the following issues:

- The title "PRMT5-dependent SRSF1 modification" is unclear. Does this mean the PRMT5-dependent interaction of SRSF1 and YTHDC1? I am not sure if the importance of this interaction in Tip60 exon inclusion is proven while SRSF1-mediated rescue of the exon inclusion is clearly presented.
- Overall, the result section contains excess discussion besides the results. Too much speculation should be avoided in the results section.
- On p. 7, they state that "a spliced Tip60 isoform, presumed to be catalytically inactive, is produced upon PRMT5 inhibition". A previous report showed that there are two Tip60 isoforms, alpha (larger) and beta (smaller) (Cell report, Cell Rep. 2018 Sep 4;24(10):2643-2657). The shorter Tip60 beta form is catalytically inactive and results from exon 5 skipping. It is unclear whether the two isoforms identified by the authors correspond to the previously defined alpha and beta isoforms. Please clarify.
- The GSE in figure 2C shows the downregulation E2F targets and G2/M checkpoint genes. These genes are known to be activated by virus T antigen oncogene. Does PRMT5 inhibitor affect alternative splicing of MCPyV T antigen transcripts?
- In Figure 1D (Max and EP400 IPs), the authors would like to include a longer exposure image to show full-length Tip60 incorporation into the EP400-Max complex. It would be expected that full-length Tip60 incorporation would be reduced by the PRMT5 inhibitor.
- In Figure EV4A, many isoforms are detected in hnRNP genes. Is the splicing of hnRNP genes altered by PRMT5 inhibition? You stated that the majority of isoforms were not significantly altered in PRMT5 inhibited cells.

Minor comments:

- On page 7, the figures are misquoted. It should read "an effect not observed at 111 nM (Figure 1C) or in the context of Dox-induced EP400 knockdown (Figure 1D). However, full-length EP400 protein levels were not significantly altered by PRMT5 inhibition in MKL-1 cells (Figure 1D).
- The Results section for the PacBio sequencing includes details of the analysis, which can be moved to the Materials and Methods section.

Response to reviewers – “PRMT5-Dependent SRSF1 Activity Promotes Tip60-EP400 Complex Activity via Exon Inclusion”

Overall Editor Comments:

The revised manuscript must include data on the reproducibility of the rescue of splicing defects by low-dose SRSF1 over expression (as suggested by Reviewer 1). Additionally, the results must be further elaborated upon, with changes to the text especially when referring to Tip60 isoforms described in this work (Reviewer 1, point 12) and in the context of published literature (Reviewer 2, second para, point 3).

○ Response:

We have calculated statistical significance based on three independent experiments for each Tip60 blot in Fig. 5A, as suggested by Reviewer 1, to demonstrate the reproducibility of SRSF1 overexpression rescuing the Tip60 α splicing defect induced by low-dose PRMT5 inhibitor treatment. The bar graphs in Fig. 5A have been updated accordingly.

We have revised the text to further clarify the effects of PRMT5 inhibition on Tip60 isoform splicing. In response to Reviewer 2 (point 3), we explicitly state that our long-read and short-read sequencing data confirm Tip60 exon 5 skipping in JNJ-64619178-treated MCC cells (Fig. 4B, C), and the resulting larger and smaller isoforms correspond to the Tip60 alpha and beta isoforms previously described by Hamard et al. in PRMT5 knockout (KO) hematopoietic stem and progenitor cells (HSPCs).

To address Reviewer 1 (point 12), noting that exon 5 does not include the catalytic domain, we revised the description of Tip60 β from “catalytically inactive” to “catalytically impaired.” We cite Hamard et al., who showed that PRMT5 KO cells predominantly express Tip60 β , the exon 5-skipped isoform, resulting in reduced acetyltransferase activity and decreased histone H4 acetylation, consistent with our observations in PRMT5-inhibited MCC cells (Fig. EV1G, H). Hamard et al. further explained that exon 5 skipping indirectly reduces Tip60 catalytic activity by removing a proline-rich region critical for protein-protein interactions, thereby impairing its chromatin-modifying function during DNA repair.

Reviewer #1 Comments:

The authors demonstrate that the MCC cell line MKL-1 is sensitive to PRMT5 inhibition, leading to reduced expression of the full-length Tip60 isoform. To explore the impact of PRMT5 inhibition on splicing, they conducted both short- and long-read sequencing, comparing untreated cells with those treated with the PRMT5 inhibitor JNJ-64619178. Their analysis revealed changes in alternative splicing, particularly in intron retention and exon skipping. The authors propose a connection between Tip60 splicing changes, the function of SRSF1 in Tip60 splicing, and the potential link of SRSF1 function to m6A RNA modification.

While the findings reinforce/confirm previously established mechanisms, they offer limited novelty. Differential gene expression and splicing analysis, the impact on Tip60 splicing and cell proliferation, the dynamics of SDMA response to inhibition, and the types of alternative splicing events observed have all been reported in multiple prior studies (Braun et al., Radziskeuskaya et al., Bezzi et al., Hamard et al., Clarke et al., and others). While reproducibility in a distinct cellular context is valuable, the experimental outcomes largely confirm existing literature rather than presenting new insights. Furthermore, the established role of PRMT5 in regulating SRSF1 methylation and its impact on splicing has been documented since 2019

(<https://www.nature.com/articles/s41594-019-0313-z>), yet this study is not cited or mentioned in the manuscript.

○ Response:

We thank the reviewer for noting the omission. The reference to Radziskeuskaya et al. 2019 and the other relevant studies have now been added in the appropriate sections of the manuscript.

Reviewer comment:

The only novel aspect of the study appears in Figures 5A and 5C but needs to be developed considerably to establish its reproducibility and relevance. The partial rescue of Tip60 splicing defects by low-dose SRSF1 overexpression may provide a mechanistic link supporting the proposed model, but *this experiment needs to be reproduced and assessed statistically*.

○ Response:

We agree that statistical validation was necessary. We have therefore repeated the SRSF1-overexpression rescue experiments and quantified full-length Tip60 α levels for each blot. The updated Fig. 5A now displays mean \pm SD ($n = 3$) with paired two-tailed t-test P-values. With a sub-optimal dose of PRMT5 inhibitor (1.37 nM JNJ-64619178), SRSF1 overexpression results in a significant, partial rescue of Tip60 α exon 5 inclusion ($P < 0.05$). At the optimal inhibitor dose (12 nM), however, no rescue is observed ($P > 0.05$). These data demonstrate that the rescue is reproducible yet dose-dependent, reinforcing the mechanistic link between PRMT5-regulated SRSF1 activity and Tip60 splicing. The bar graph and legend in Fig. 5A have been revised accordingly.

Reviewer comment:

The potential involvement of m6A methylation and YTHDC1 is intriguing but remains unexplored by the authors. Further mechanistic studies, such as assessing whether an unmethylatable SRSF1 mutant can rescue splicing defects or bind to YTHDC1, would enhance the study's contribution beyond existing knowledge.

○ Response:

We greatly appreciate Reviewer 1's insightful suggestion to explore the involvement of m6A methylation and YTHDC1 interactions further using an unmethylatable SRSF1 mutant. While

we agree that these experiments could enhance the manuscript's impact and originality, they would require significant additional time beyond the 3-month revision window provided. We therefore highlight this excellent suggestion explicitly in our Discussion section as a promising direction for future mechanistic studies.

Additional Comments:

1. The manuscript states that 68.82% of identified isoforms were classified as novel. Is this proportion typical for long-read RNA sequencing studies?
 - Response: Yes, the high proportion of novel isoforms is consistent with other studies using PacBio Iso-Seq. We have added references for comparison, including Wang et al. (2024), who reported 82.9% novel isoforms in human tissues, and Glinos et al. (2022), who reported approximately 77%.
2. Do the genes exhibiting differential isoform expression overlap with those identified as differentially expressed?
 - Response: We appreciate this question and have expanded our analysis to include a direct comparison between isoform-level and gene-level differential expression patterns, rather than relying solely on gene-level aggregated data for the comparison. A new paragraph has been added to the Results section under the heading 'Full-length isoform (PacBio Iso-Seq) Gene Expression Levels Correlate With Those of Short-read (Illumina RNA-Seq) Datasets' to describe this comparison. Specifically, we compared genes with at least one differentially expressed isoform in the PacBio Iso-Seq dataset to the genes identified as differentially expressed at the gene level in the Illumina RNA-seq dataset. Of the 3,092 genes with at least one differentially expressed isoform, 1,735 overlapped with the 5,394 differentially expressed genes from the Illumina data. This indicates a large portion of isoform-level expression changes (1,357 genes) occur without significant changes in overall gene-level abundance, suggesting regulation at the level of isoform usage. Conversely, 3,659 genes identified as differentially expressed in the Illumina dataset did not show isoform-level changes in the PacBio data.
3. The manuscript should clarify the classification of isoforms (e.g., novel in catalog vs. novel not in catalog). Similarly, the categories in Figure 3A require explanation.
 - Response: We thank the reviewer for this suggestion and have clarified the isoform classification system in the manuscript by further describing the SQANTI framework (<https://github.com/ConesaLab/SQANTI3/wiki/SQANTI3-isoform-classification:-categories-and-subcategories>). These descriptions are added to the Methods section, and we have expanded the Figure 3A legend to define the structural categories shown in the plot.
4. In EV3, the cut-off values for differentially expressed genes (DEGs) should be included in the figure legend.
 - Response: We have added DEG cutoff values to the legend for figure EV3.
5. The following paragraph seems incomplete. It should explicitly state that these conditions were used for subsequent experiments: "To ensure that the observed effects were specifically due to PRMT5 inhibition and not secondary to cell death, we established

experimental conditions in which MKL-1 cells were treated with 12 nM JNJ-64619178 for fewer than five days. Under these conditions, any changes detected could be attributed to PRMT5 inhibition rather than cytotoxicity”

- Response: We have clarified that the 12 nM ≤5 day treatment regimen was used as the optimal concentration for downstream experiments to specifically attribute observed effects to PRMT5 inhibition rather than cytotoxicity, while concentrations lower than 12 nM were defined as sub-optimal.
6. A figure reference is missing in this statement: “In an MKL-1 stable cell line expressing doxycycline (Dox)-inducible EP400 shRNA, treatment with either Dox or 12 nM JNJ-64619178 resulted in reduced acetylation of histone H4 at lysine 5, 12, and 16, as well as decreased acetylation of the H2A.Z histone variant”
- Response: We have added the figure reference (Fig. EV1G, H) to this statement.
7. The manuscript states: "However, full-length EP400 protein levels were not significantly altered by PRMT5 inhibition in MKL-1 cells" Where is the supporting evidence for this claim?
- Response: We have added a figure reference (Fig. 1D) to this statement.
8. The following paragraph belongs in the discussion section rather than the results: "P53 is the most frequently mutated tumor suppressor in human cancer. Several tumor viruses, including polyomaviruses, papillomaviruses, and adenoviruses, express viral tumor antigens that bind to and inactivate p53... Therefore, it would be important to investigate in future studies whether inhibiting p53 activation can desensitize MCC cells to PRMT5 deficiency or if overexpressing MYC is synergistically lethal with PRMT5 inhibition."
- Response: We thank the reviewer for pointing this out. We have moved the paragraph from the Results to the Discussion to keep interpretative material separate from descriptive findings.
9. Figure 2B presents selected gene names. How were these genes chosen for display?
- Response: Figure 2B was generated using the EnhancedVolcano package in R, which automatically selects and labels genes based on statistical significance and effect size. Genes with adjusted p-values < 0.05 and absolute log2 fold change > 1 were considered and no manual curation of gene labels was performed. EnhancedVolcano automatically displays as many top-ranked genes as possible within the available plot space while minimizing label overlap and visual clutter. This information is added to the figure legend.
10. The logic in the following statement is unclear: "The substantial presence of large numbers of hnRNP isoforms in MCC cells suggests an important regulatory balance involving SRSF proteins and their post-translational modifications, such as phosphorylation or PRMT-mediated methylation." A large number of isoforms does not inherently imply a regulatory balance between PTMs. This reasoning should be better articulated.
- Response: We appreciate the reviewer’s point regarding clarity. We have revised the text to explicitly articulate the logic: extensive hnRNP isoform diversity indicates frequent alternative splicing of hnRNP transcripts themselves, reflecting an active, PTM-regulated competition between SRSF proteins and hnRNPs. Under PRMT5 deficiency, loss of

SDMA marks weakens specific interactions between these protein families, disrupting their competitive balance and contributing to the selective splicing defects observed in our study.

11. The following statement lacks clarity: "The mix of upregulated (red) and downregulated (blue) genes suggests that PRMT5 inhibition influences alternative splicing patterns." How does a mixture of up- and downregulated genes imply a specific effect on alternative splicing?
 - Response: We thank the reviewer for pointing this out. We agree that the original statement was unclear. In response, we have revised the text to clarify that it is not the presence of both up- and downregulated isoforms per se that suggests a splicing effect, but rather the distribution of regulation across specific isoform structural categories. We now highlight that the chi-squared test revealed significant differences in the abundance of up- and downregulated isoforms across splicing-related categories (e.g., intron retention, exon skipping, novel splice sites), which provides stronger evidence that PRMT5 inhibition disrupts alternative splicing.
12. Figure 4C highlights the skipped exon of Tip60, but this exon does not include the catalytic domain. The manuscript previously stated that loss of the full-length Tip60 isoform leads to a catalytically inactive protein. This apparent contradiction should be addressed.
 - Response: We agree that exon 5 does not encode residues within the Tip60 catalytic (MYST) domain. We have therefore replaced the phrase "catalytically inactive" with "catalytically impaired" throughout the manuscript. Hamard *et al.* (2018) demonstrated that PRMT5-knockout HSPCs predominantly express Tip60 β (the exon-5-skipped isoform), whose loss of a 54-aa proline-rich segment diminishes auto-acetylation and weakens protein-protein interactions, leading to reduced HAT activity and lower histone-H4 acetylation. We now cite this study and note that we observe the same decrease in H4 acetylation under PRMT5 inhibition in MCC cells (Fig. EV1G,H), reinforcing that exon 5 skipping impairs, rather than abolishes, Tip60's enzymatic function.
13. Figure 4D's legend should provide a more detailed explanation of the graph and the meaning of different lines.
 - Response: We agree that the original legend lacked sufficient detail. The legend for Fig. 4D has been rewritten in full to describe the axes, trace types, statistical test, and biological interpretation.

Reviewer conclusion:

This study reinforces known mechanisms of PRMT5-mediated splicing regulation but offers limited new insights. The findings in Figures 5A and 5C provide the most meaningful additions, yet their mechanistic underpinnings remain underexplored. Addressing the questions posed above would enhance the impact and originality of the work.

Reviewer #2:

The manuscript by Sevigny et al. demonstrated that PRMT5, which regulates splicing of Tip60/KAT5, is required for Merkel cell polyomavirus (MCPyV)-positive Merkel cell carcinoma (MCC) tumorigenicity. Dr. Cheng, a corresponding author of this manuscript, previously identified the MycL/Max-EP400-Tip60 histone acetyltransferase complex as one of the critical cellular oncogenic targets of the Merkel cell polyomavirus (MCPyV) small T oncoprotein (ST). In this study, his group found that one of the Tip60 isoforms, which appears to be catalytically inactive and results from exon 5 skipping, is increased by PRMT5 inhibitors. PRMT5 inhibition suppressed histone acetylation associated with the Tip60-HAT complex and suppressed proliferation of virus-positive MCC tumor cells. In addition, the authors identified SRSF1 as a regulator of Tip60 mRNA. Overall, the experiments are carefully performed and the results support their conclusion.

However, the manuscript can be improved if the authors address the following issues:

1. The title "PRMT5-dependent SRSF1 modification" is unclear. Does this mean the PRMT5-dependent interaction of SRSF1 and YTHDC1? I am not sure if the importance of this interaction in Tip60 exon inclusion is proven while SRSF1-mediated rescue of the exon inclusion is clearly presented.
 - Response: We thank the reviewer for highlighting the ambiguity in the original title and the mechanistic link between PRMT5, SRSF1, and Tip60 exon 5 inclusion. We have changed the title from "PRMT5-dependent SRSF1 modification" to "PRMT5-dependent SRSF1 activity." In the Introduction we now cite Radzishchanskaya *et al.* 2019, which biochemically demonstrated that PRMT5 catalyzes symmetric dimethyl-arginine (SDMA) on SRSF1. This provides the documented basis for our working model. We propose that PRMT5-catalyzed SDMA of SRSF1 is required for its interaction with YTHDC1 (Fig. 5C), and that this interaction is, in turn, necessary for efficient Tip60 exon 5 inclusion. Figure 5A now displays mean \pm SD from three independent experiments ($n = 3$) with *t*-test *P*-values. The revised data show that HA-SRSF1 partially but significantly rescues Tip60 α exon 5 inclusion at a sub-optimal PRMT5-inhibitor dose (1.37 nM JNJ-64619178; $P < 0.05$) but fails to rescue at the optimal dose (12 nM). These results underscore a dose-dependent requirement for PRMT5-regulated SRSF1 activity: when PRMT5 is largely inhibited (12 nM), simply increasing SRSF1 abundance cannot compensate, consistent with the need for PRMT5-mediated SDMA to fully support Tip60 exon 5 inclusion.
2. Overall, the result section contains excess discussion besides the results. Too much speculation should be avoided in the results section.
 - Response: We thank the reviewer for highlighting this point. We have systematically separated data presentation from interpretation by trimming the Results to observation-only statements and relocating speculative or forward-looking sentences from the Results to the Discussion.
3. On p. 7, they state that "a spliced Tip60 isoform, presumed to be catalytically inactive, is produced upon PRMT5 inhibition". A previous report showed that there are two Tip60 isoforms, alpha (larger) and beta (smaller) (Cell report, Cell Rep. 2018 Sep 4;24(10):2643-2657). The shorter Tip60 beta form is catalytically inactive and results from exon 5 skipping. It is unclear whether the two isoforms identified by the authors correspond to the previously defined alpha and beta isoforms. Please clarify.

- Response: We appreciate the request for clarification. The manuscript now explicitly links the two Tip60 bands we detect to the previously characterized Tip60 α and Tip60 β isoforms reported by Hamard et al. (2018, *Cell Reports*).
4. The GSE in figure 2C shows the downregulation E2F targets and G2/M checkpoint genes. These genes are known to be activated by virus T antigen oncogene. Does PRMT5 inhibitor affect alternative splicing of MCPyV T antigen transcripts?
- Response: We thank the reviewer for pointing out the significance of these findings. To address this, we examined the expression and alternative splicing patterns of the small and large T antigen transcripts from (MCPyV) in our dataset. We did not observe significant differences (adjusted p-value < 0.05) in the expression or splicing of MCPyV T antigen transcripts between PRMT5-inhibited and control samples. These results are now provided in a new supplementary table (Table S9) and described in the Results section of the manuscript.
5. In Figure 1D (Max and EP400 IPs), the authors would like to include a longer exposure image to show full-length Tip60 incorporation into the EP400-Max complex. It would be expected that full-length Tip60 incorporation would be reduced by the PRMT5 inhibitor.
- Response: We appreciate this concern and have now provided a longer exposure Western blot to more clearly show full-length Tip60 (Tip60 α) incorporation into the EP400-Max complex. Due to the predominance of the Tip60 β isoform in the EP400-Max complex, only a very small fraction of Tip60 α is detectable even under these improved conditions. Given these inherently low basal levels of Tip60 α , subtle changes in its abundance are likely to substantially influence EP400-Max complex activity. Although IP-Western analysis is insufficiently sensitive to reliably detect a clear reduction in Tip60 α incorporation after PRMT5 inhibition, we do observe a robust increase in Tip60 β incorporation into the complex. We propose that increased Tip60 β competes with Tip60 α for binding to the EP400-Max complex, thus indirectly reducing the incorporation of Tip60 α .
6. In Figure EV4A, many isoforms are detected in hnRNP genes. Is the splicing of hnRNP genes altered by PRMT5 inhibition? You stated that the majority of isoforms were not significantly altered in PRMT5 inhibited cells.
- Response: We appreciate the reviewer's observation regarding isoform detection in hnRNP genes (Figure EV4A). To clarify, three of the twenty-nine annotated hnRNP genes (HNRNPUL2-BSCL2, HNRNPL, and HNRNPH1) were identified as differentially expressed at the gene level (adjusted p < 0.05; see Table S2). Across all hnRNP genes, we detected a total of 865 isoforms, with isoform counts per gene ranging from 1 to 146. Among these, 80 isoforms from 11 unique hnRNP genes were significantly differentially expressed (adjusted p < 0.05). The most pronounced isoform-level changes were observed in HNRNPK and HNRNPC, which had 33 and 24 differentially expressed isoforms, respectively. These findings indicate that while the majority of hnRNP gene isoforms were not significantly altered, multiple isoforms from specific genes were affected by PRMT5 inhibition. We have added this information to the Results under section 'High hnRNP isoform numbers imply SRSF-hnRNP antagonism in fine-tuning alternative splicing' paragraph two. We also updated Figure EV4 to include details of the differential expression results.

7. Minor comments:

On page 7, the figures are misquoted. It should read "an effect not observed at 111 nM (Figure 1C) or in the context of Dox-induced EP400 knockdown (Figure 1D). However, full-length EP400 protein levels were not significantly altered by PRMT5 inhibition in MKL-1 cells (Figure 1D).

- Response: We thank the reviewer for identifying this error. The manuscript has now been corrected accordingly.

The Results section for the PacBio sequencing includes details of the analysis, which can be moved to the Materials and Methods section.

- Response: We thank the reviewer for pointing this out and have moved the details of the PacBio analysis to the Methods section.

July 31, 2025

Re: Life Science Alliance manuscript #LSA-2025-03316-TR

Dr. Jingwei Cheng
University of New Hampshire
Molecular, Cellular and Biomedical Sciences
46 College Road
Rudman 206
Durham, NH 03824

Dear Dr. Cheng,

Thank you for submitting your revised manuscript entitled "PRMT5-Dependent SRSF1 Activity Promotes Tip60-EP400 Complex Activity via Exon Inclusion" to Life Science Alliance. The manuscript has been seen by the original reviewers whose comments are appended below. While the reviewers continue to be overall positive about the work in terms of its suitability for Life Science Alliance, some important issues remain.

We request you to follow the request from Reviewer 1 related to quantification of Western blots, a clear indication of number of replicates for each experiment, and the provision of all uncropped Western blot images (including replicates) as supplementary material. We agree with the reviewer that this supplementary information, in particular for Figure 5, would be important to establish the advance described in this work.

Our general policy is that papers are considered through only one revision cycle; however, given that the suggested changes are relatively minor, we are open to one additional short round of revision. Please note that I will expect to make a final decision without additional reviewer input upon re-submission.

Please submit the final revision along with a letter that includes a point by point response to the remaining reviewer comments.

B. MANUSCRIPT ORGANIZATION AND FORMATTING:

Sincerely,

Sarita Hebbar, PhD
Scientific Editor
Life Science Alliance
<http://www.lsajournal.org>

Reviewer #1 (Comments to the Authors (Required)):

The authors have addressed the majority of my comments and significantly improved the manuscript.

My main point was regarding the replication of the key experiment in Figure 5A, which they did. However, they did not provide Western blot images for independent replications. Please provide all the images as supplementary materials.

Additionally, since the manuscript contains multiple Western blots, I recommend that all the images should be:

- a) quantified as in Figure 5A and there should be a statement how many times the westerns were repeated (quantifications can be provided in Supplementary Materials)
- b) provided also as uncropped blots to increase transparency (in Supplementary Materials).

Reviewer #2 (Comments to the Authors (Required)):

The authors addressed all the concerns, and the revised manuscript was significantly improved.

Response to reviewers – “PRMT5-Dependent SRSF1 Activity Promotes Tip60-EP400 Complex Activity via Exon Inclusion”

Overall Editor Comments:

The manuscript has been seen by the original reviewers whose comments are appended below. While the reviewers continue to be overall positive about the work in terms of its suitability for Life Science Alliance, some important issues remain. We request you to follow the request from Reviewer 1 related to quantification of Western blots, a clear indication of number of replicates for each experiment, and the provision of all uncropped Western blot images (including replicates) as supplementary material. We agree with the reviewer that this supplementary information, in particular for Figure 5, would be important to establish the advance described in this work.

- Response: We thank the editor for summarizing the reviewers' comments. As requested, we have carefully addressed all of Reviewer #1's points regarding Western blot quantification, replicate clarification, and provision of uncropped images. Specific responses and supplementary files addressing these concerns are detailed under Reviewer #1's comments.

Reviewer #1 Comments:

The authors have addressed the majority of my comments and significantly improved the manuscript. My main point was regarding the replication of the key experiment in Figure 5A, which they did. However, they did not provide Western blot images for independent replications. Please provide all the images as supplementary materials. Additionally, since the manuscript contains multiple Western blots, I recommend that all the images should be:

- a) quantified as in Figure 5A and there should be a statement how many times the westerns were repeated (quantifications can be provided in Supplementary Materials)
 - b) provided also as uncropped blots to increase transparency (in Supplementary Materials).
- Response: We appreciate reviewer #1's thorough evaluation of our manuscript. In response, we have:
 - Included all uncropped Western blot images, including replicates, in the supplementary file "**Uncropped Western Blots.pptx.**"
 - Provided quantifications for all Western blots in the supplementary file "**Uncropped Western Blots.pptx.**"
 - Clearly indicated the number of replicates for each Western blot in the supplementary file "**Western Blot Replicate Summary.docx.**"

Reviewer #2 Comments:

The authors addressed all the concerns, and the revised manuscript was significantly improved.

- Response: We thank Reviewer #2 for carefully reviewing the revised manuscript and appreciate the positive assessment of our improvements.

August 8, 2025

RE: Life Science Alliance Manuscript #LSA-2025-03316-TRR

Dr. Jingwei Cheng
University of New Hampshire
Molecular, Cellular and Biomedical Sciences
46 College Road
Rudman 206
Durham, NH 03824

Dear Dr. Cheng,

Thank you for submitting your revised manuscript entitled "PRMT5-Dependent SRSF1 Activity Promotes Tip60-EP400 Complex Activity via Exon Inclusion". We would be happy to publish your paper in Life Science Alliance pending final revisions necessary to meet our formatting guidelines.

- Please add the molecular weight markers to blot images in Figure 5B and C.
- We encourage you to edit your title to make it informative without the use of too many abbreviations.
- Please provide necessary consent and ethical statement for the photograph of patient in Figure 6.
- In the methods section, please provide the details on:
 - all cell lines used (citation, passage number, source).
 - all antibodies used (source, concentration).
 - quantification of immunoblots and information on reference protein/house keeping protein.
- Please include a statement in the methods section that information on, and images of, immunoblot replicates are provided as a supplemental data file.
- Please add callouts for Figures 6A, 6B, S4A, and supplemental data file on immunoblot replicates, to your main manuscript text.
- Please add your main, supplementary figure, and table legends to the main manuscript text after the references section and remove them from the figures.
- Please provide one main figure per page.
- LSA allows supplementary figures, but no EV Figures; please update your callouts for the Supplementary Figures in the manuscript Fig EV1A=Fig S1A; while supplementary figures use the system supplementary Fig S1.
- Please upload your Table in editable .doc or excel format.
- Please add the X and Bluesky handles of your host institute/organisation as well as your own and/or one of the authors in our system.
- Please be sure that the authorship listing and order are correct.

LSA now encourages authors to provide a 30-60 second video where the study is briefly explained. We will use these videos on social media to promote the published paper and the presenting author (for examples, see <https://docs.google.com/document/d/1-UWCfbE4pGcDdcgzcmiuJI2XMBJnxKYeqRvLLrLS0s8s/edit?usp=sharing>). Corresponding or first-authors are welcome to submit the video. Please submit only one video per manuscript. The video can be emailed to contact@life-science-alliance.org

A. FINAL FILES:

B. MANUSCRIPT ORGANIZATION AND FORMATTING:

Sincerely,

Sarita Hebbar, PhD
Scientific Editor
Life Science Alliance
<http://www.lsajournal.org>

Response to editors – “PRMT5-Dependent SRSF1 Activity Promotes Tip60-EP400 Complex Activity via Exon Inclusion”

Editor Comments:

-Please add the molecular weight markers to blot images in Figure 5B and C.

- Response: We have added molecular weight markers to the immunoblot images in Figure 5B and Figure 5C, as requested.

-We encourage you to edit your title to make it informative without the use of too many abbreviations.

- Response: We have revised the title to reduce abbreviations and improve clarity. The new title of the manuscript is: “Protein Arginine Methyltransferase 5 Sustains Tip60-EP400 Complex via SRSF1 in Merkel Cell Carcinoma”. This title spells out “Protein Arginine Methyltransferase 5” (for PRMT5).

-Please provide necessary consent and ethical statement for the photograph of patient in Figure 6.

- Response: We removed the external patient photograph from Figure 6 and replaced it with the text ‘Merkel cell carcinoma’. No patient image remains; consent is not applicable.

-In the methods section, please provide the details on:

--all cell lines used (citation, passage number, source).

--all antibodies used (source, concentration).

--quantification of immunoblots and information on reference protein/house keeping protein.

- Response: We revised Methods to add cell line sources, passage ranges, a table of antibodies with vendor and dilution, and immunoblot quantification.

-Please include a statement in the methods section that information on, and images of, immunoblot replicates are provided as a supplemental data file.

- Response: We added a statement in Methods Immunoblot Analysis section that uncropped and replicate immunoblot images are provided in a Supplementary Data file.

-Please add callouts for Figures 6A, 6B, S4A, and supplemental data file on immunoblot replicates, to your main manuscript text.

- Response: We added text references to Figures 6A, 6B, S4A, and to the Supplementary Data file with the immunoblot replicates.

-Please add your main, supplementary figure, and table legends to the main manuscript text after the references section and remove them from the figures.

- Response: We moved all main, supplementary figure, and table legends to the manuscript after the References.

-Please provide one main figure per page.

- Response: We reformatted the figures so each main figure occupies its own page.

-LSA allows supplementary figures, but no EV Figures; please update your callouts for the Supplementary Figures in the manuscript Fig EV1A=Fig S1A; while supplementary figures use the system supplementary Fig S1.

- Response: We converted all EV callouts to the Supplementary Figure format (e.g., Fig EV1A → Fig S1A)

-Please upload your Table in editable .doc or excel format.

- Response: We uploaded all tables as editable .xlsx files.

-Please add the X and Bluesky handles of your host institute/organisation as well as your own and/or one of the authors in our system.

- Response: We could not find X and Bluesky fields in the revision portal. We list them here:
Institution X: @UofNH
Institution Bluesky: @unhresearch.bsky.social
Corresponding author X: @jingweicheng2
Corresponding author Bluesky: @JingweiCheng.bsky.social

-Please be sure that the authorship listing and order are correct.

- Response: We confirm the authorship listing and order are correct.

August 12, 2025

RE: Life Science Alliance Manuscript #LSA-2025-03316-TRRR

Dr. Jingwei Cheng
University of New Hampshire
Molecular, Cellular and Biomedical Sciences
46 College Road
Rudman 206
Durham, NH 03824

Dear Dr. Cheng,

Thank you for submitting your Research Article entitled "Protein Arginine Methyltransferase 5 Sustains Tip60-EP400 Complex via SRSF1 in Merkel Cell Carcinoma". It is a pleasure to let you know that your manuscript is now accepted for publication in Life Science Alliance. Congratulations on this interesting work.

DISTRIBUTION OF MATERIALS:

Again, congratulations on a very nice paper. I hope you found the review process to be constructive and are pleased with how the manuscript was handled editorially. We look forward to future exciting submissions from your lab.

Sincerely,

Sarita Hebbar, PhD
Scientific Editor
Life Science Alliance
<http://www.lsajournal.org>